# Learning with Long-term Remembering: Following the Lead of Mixed Stochastic Gradient

## Abstract

Current deep neural networks can achieve remarkable performance on a single task. However, when the deep neural network is continually trained on a sequence of tasks, it seems to gradually forget the previous learned knowledge. This phenomenon is referred to as *catastrophic forgetting* and motivates the field called lifelong learning. The central question in lifelong learning is how to enable deep neural networks to maintain performance on old tasks while learning a new task. In this paper, we introduce a novel and effective lifelong learning algorithm, called MixEd stochastic GrAdient (MEGA), which allows deep neural networks to acquire the ability of retaining performance on old tasks while learning new tasks. MEGA modulates the balance between old tasks and the new task by integrating the current gradient with the gradient computed on a small reference *episodic memory*. Extensive experimental results show that the proposed MEGA algorithm significantly advances the state-of-the-art on all four commonly used lifelong learning benchmarks, reducing the error by up to 18%.

## 1 Introduction

A significant step towards artificial general intelligence (AGI) is to enable the learning agent to acquire the ability of remembering past experiences while being trained on a continuum of tasks. Current deep neural networks are capable of achieving remarkable performance on a single task (Goodfellow et al., 2016). However when the network is retrained on a new task, its performance drops drastically on previously trained tasks, a phenomenon which is referred to as *catastrophic forgetting* (Ratcliff, 1990; Robins, 1995; French, 1999; Kirkpatrick et al., 2017). In stark contrast, human cognitive system is capable of acquiring new knowledge without damaging previously learned experiences. It is thus of great importance to develop algorithms to allow deep neural networks to achieve continual learning capability (i.e., avoiding catastrophic forgetting).

The problem of catastrophic forgetting motivates the field called lifelong learning (Thrun & Mitchell, 1995; Kirkpatrick et al., 2017; Parisi et al., 2019). A central dilemma in lifelong learning is how to achieve a balance between the performance on old tasks and the new task (Robins, 1995; Kirkpatrick et al., 2017). During the process of learning the new task, the originally learned knowledge will typically be disrupted, which leads to catastrophic forgetting. On the other hand, a learning algorithm biasing towards old tasks will interfere with the learning of the new task. Several lines of methods are proposed recently to address this issue. Examples include regularization based methods (Kirkpatrick et al., 2017; Zenke et al., 2017), knowledge transfer based methods (Rusu et al., 2016), episodic memory based methods (Lopez-Paz et al., 2017; Chaudhry et al., 2018b; Riemer et al., 2018). However, the existing methods require over-parameterized neural networks (Kirkpatrick et al., 2017; Chaudhry et al., 2018a) or are not flexible enough handle the stochastic nature of the learning process (Lopez-Paz et al., 2017; Chaudhry et al., 2018b) since they did not explicitly consider the model's performance on the current task and old tasks in the learning process.

In this paper, we propose a novel and effective lifelong learning algorithm, called MixEd stochastic GrAdient (MEGA), to address the catastrophic forgetting problem. We cast the problem of balancing the performance on old tasks and the new task as an optimization problem with composite objective. Our formulation is general and closely related to several recently proposed lifelong learning

algorithms (Lopez-Paz et al., 2017; Chaudhry et al., 2018b; Riemer et al., 2018). We approximately solve the optimization problem using one-step stochastic gradient descent with the standard gradient replaced by the proposed mixed stochastic gradient. The mixed stochastic gradient is derived from the gradients computed on the data of the current task and an episodic memory which stores a small subset of observed examples from old tasks (Lopez-Paz et al., 2017; Chaudhry et al., 2018b; Riemer et al., 2018). Based on our derivation, the direction of the mixed stochastic gradient balances the loss on old tasks and the new task in an adaptive manner. Therefore, the proposed MEGA algorithm allows deep neural networks to learn new tasks while avoiding catastrophic forgetting.

Our contributions are as follows. (1) We propose a novel and effective algorithm, called MEGA, for lifelong learning problems. (2) We extensively evaluate our algorithm using conventional lifelong learning benchmark datasets, and the results show that the proposed MEGA algorithm significantly advances the state-of-the-art performance across all the datasets. MEGA achieves an average accuracy of 91.21±0.10% on **Permuted MNIST**, which is 2% better than the previous state-of-the-art model. On **Split CIFAR**, our proposed MEGA achieves an average accuracy of 66.12±1.93%, which is about 5% better than the state-of-the-art method. Specially, on the **Split CUB** dataset, MEGA achieves an average accuracy of 80.58±1.94%, which surpasses the multi-task baseline which is previously believed as an upper bound performance of lifelong learning algorithms (Chaudhry et al., 2018b). (3) Finally, we also show that the proposed MEGA algorithm can handle increasingly non-stationary settings when the number of tasks becomes significantly larger.

## 2 RELATED WORK

Improving the continual learning ability of neural network is a prerequisite to extending it to more practical vision tasks. Several lines of lifelong learning methods are proposed recently, we categorize them into different types based on the methodology,

**Regularization based approaches:** EWC (Kirkpatrick et al., 2017) adopts Fisher information matrix to prevent important weights for old tasks from changing drastically. In PI (Zenke et al., 2017), the authors introduce *intelligent synapses* which endows each individual synapse with a local measure of "importance" to avoid old memories from being overwritten. RWALK (Chaudhry et al., 2018a) utilizes a KL-divergence based regularization for preserving knowledge of old tasks. While in MAS (Aljundi et al., 2018) the importance measure for each parameter of the network is computed based on how sensitive the predicted output function is to a change in this parameter.

**Knowledge transfer based methods:** PROG-NN (Rusu et al., 2016) is a representative knowledge transfer based lifelong method. In PROG-NN, a new "column" with lateral connections to previous hidden layers is added for each new task. The lateral connections allow the new task to leverage the knowledge extracted from the old tasks. Recently, in Lee et al. (2019), the authors proposed a method to leverage unlabeled data in the wild to avoid catastrophic forgetting using knowledge distillation. The authors introduce *global distillation* for incorporating unlabeled data in the learning process.

**Episodic memory based approaches:** In episodic memory based lifelong learning methods, a small episodic memory is used for storing a subset of the examples from old tasks. Different episodic memory based approaches differ in the way of computing the reference gradients of the episodic memory. GEM (Lopez-Paz et al., 2017) computes the reference gradients using each individual previous tasks while in AGEM (Chaudhry et al., 2018b) the reference gradient is computed on the episodic memory by randomly sampling a batch of examples. MER a (Riemer et al., 2018) is recently proposed lifelong learning algorithm which maintains an experience replay style memory with reservoir sampling and employs a meta-learning style training strategy. Several methods are proposed recently to improve the episodic memory based approaches. In (Aljundi et al., 2019), a line of methods are proposed to select important samples to store in the memory in order to reduce memory size. Instead of storing samples, in (Farajtabar et al., 2019) the authors proposed Orthogonal Gradient Descent (OGD) which projects the gradients on the new task onto a subspace in which the projected gradient will not affect the model's output on old tasks and is still useful for learning the new task.

There are also some other lifelong learning methods which do not fall in above categories. In (He & Jaeger, 2018), the authors proposed a variant of the standard back-propagation algorithm

called conceptor aided backprop that shields gradients against degradation of performance on old task. Zeng et al. (2019) proposed orthogonal weights modification (OWM) to enables networks to continually learn different mapping rules in a context-dependent way.

Our proposed method belongs to episodic memory based approach and is most related to Chaudhry et al. (2018b). Our work differs from Chaudhry et al. (2018b) in two aspects. First, we introduce a more effective to modify the direction of the current gradient which put equal emphasis on both old tasks and the new task. Second, we explicitly consider the performance of the model on old tasks and the new task in the process of modifying the gradient direction. Our method is also related to several multi-task learning works (Sener & Koltun, 2018; Kendall et al., 2018; Chen et al., 2017). In (Sener & Koltun, 2018; Kendall et al., 2018), the authors aim at achieving a good balance between different tasks by learning to weigh the loss on each task . In contrast, our approach directly leverages loss information in the context of lifelong learning for overcoming catastrophic forgetting. Compared with (Chen et al., 2017), instead of using the gradient norm information, our method and Lopez-Paz et al. (2017); Chaudhry et al. (2018b) focus on modifying the direction of the current gradient.

## 3 LIFELONG LEARNING

### 3.1 PROBLEM STATEMENT

Lifelong learning (LLL) (Rusu et al., 2016; Kirkpatrick et al., 2017; Lopez-Paz et al., 2017; Chaudhry et al., 2018b) considers the problem of learning a new task without degrading performance on old tasks, i.e., to avoid *catastrophic forgetting* (French, 1999; Kirkpatrick et al., 2017). Suppose there are $T$ tasks which are characterized by $T$ datasets: $\{D_1, D_2, .., D_T\}$. Each dataset $D_t$ consists of a list of triplets $(x_i, y_i, t)$, where $y_i$ is the label of $i$-th example $x_i$, and $t$ is a task descriptor that indicates which task the example comes from. Similar to supervised learning, each dataset $D_t$ is split into a training set $D_t^{tr}$ and a test set $D_t^{te}$.

In the learning protocol introduced in Chaudhry et al. (2018b), the tasks are separated into $D^{CV} = \{D_1, D_2, ..., D_{T^{CV}}\}$ and $D^{EV} = \{D_{T^{CV}+1}, D_{T^{CV}+2}, ..., D_T\}$. $D^{CV}$ is used for cross-validation to search for hyperparameters. $D^{EV}$ is used for actual training and evaluation. As pointed out in Chaudhry et al. (2018b), some regularization-based lifelong learning algorithms, e.g., Elastic Weight Consolidation (Kirkpatrick et al., 2017), are sensitive to the choice of the regularization parameters. Introducing $D^{CV}$ can help find the best regularization parameter without exposing the actual training and evaluation data. While searching for the hyperparameters, we can have multiple passes over the examples in $D^{CV}$, the training is performed on $D^{EV}$ with only a *single* pass over the examples (Lopez-Paz et al., 2017; Chaudhry et al., 2018b).

In lifelong learning, a given model $f(x; \mathbf{w})$ is trained sequentially on a series of tasks $\{D_{T^{CV}+1}, D_{T^{CV}+2}, ..., D_T\}$. When the model $f(x; \mathbf{w})$ is trained on task $D_t$, the goal is to predict the labels of the examples in $D_t^{te}$ by minimizing the empirical loss $\ell_t(\mathbf{w})$ on $D_t^{tr}$ in an online fashion without suffering accuracy drop on $\{D_{T^{CV}+1}^{te}, D_{T^{CV}+2}^{te}, ..., D_t^{te}\}$.

### 3.2 EVALUATION METRICS

Average Accuracy and Forgetting Measure (Chaudhry et al., 2018a) are common used metrics for evaluating performance of lifelong learning algorithms. In Chaudhry et al. (2018b), the authors introduce another metric, called Learning Curve Area (LCA), to assess the learning speed of different lifelong learning algorithms. In this paper, we further introduce a new evaluation metric, called Long-term Remembering (LTR), to characterize the ability of lifelong learning algorithms for *remembering* the performance of tasks trained in the far past.

Suppose there are $M_k$ mini-batches in the training set of task $D_k$. Similar to Chaudhry et al. (2018b), we define $a_{k,i,j}$ as the accuracy on the test set of task $D_j$ after the model is trained on the $i$-th mini-batch of task $D_k$. Generally, suppose the model $f(x; \mathbf{w})$ is trained on a sequence of $T$ tasks $\{D_1, D_2, ..., D_T\}$. Average Accuracy and Forgetting Measure after the model is trained on the task $D_k$ are defined as

$$A_k = \frac{1}{k} \sum_{j=1}^{k} a_{k,M_k,j} \quad F_k = \frac{1}{k-1} \sum_{j=1}^{k-1} f_j^k \tag{1}$$

where $f_j^k = \max_{l \in \{1,2,...,k-1\}} a_{l,M_l,j} - a_{k,M_k,j}$. Clearly, $A_T$ is the average test accuracy and $F_T$ assesses the degree of accuracy drop on old tasks after the model is trained on all the $T$ tasks. Learning Curve Area (LCA) (Chaudhry et al., 2018b) at $\beta$ is defined as,

$$\text{LCA}_\beta = \frac{1}{\beta + 1} \sum_{b=0}^{\beta} Z_b \tag{2}$$

where $Z_b = \frac{1}{T} \sum_{k=1}^{T} a_{k,b,k}$. Intuitively, LCA measures the learning speed of different lifelong learning algorithms. A higher value of LCA indicates that the model learns quickly. We refer the readers to Chaudhry et al. (2018b) for more details about LCA.

All the metrics introduced above fail to capture one important aspect of lifelong learning algorithms, that is, the ability to retain performance on the tasks trained in the far past. In this paper we introduce a new metric, called Long-Term Remembering (LTR), which is defined as

$$\text{LTR} = -\frac{1}{T-1} \sum_{j=1}^{T-1} (T-j) \min\{0, a_{T,M_T,j} - a_{j,M_j,j}\} \tag{3}$$

After the model is trained on all the $T$ tasks, LTR quantifies the accuracy drop on task $D_j$ relative to $a_{j,M_j,j}$. The coefficient $T - j$ emphasizes more on the tasks trained earlier. Different algorithms can have the same average accuracy but very different LTR based on their ability to maintain the performance on the past tasks (a.k.a, *long-term remembering*).

## 4 MIXED STOCHASTIC GRADIENT

In this section, we introduce the proposed Mixed Stochastic Gradient (MEGA) algorithm. Following previous works (Lopez-Paz et al., 2017; Chaudhry et al., 2018b), when the model is trained on the $t$-th task, an episodic memory $M$ is used for storing a subset of the examples from all the old tasks $k < t$. The main idea of MEGA is to minimize the loss on the episodic memory and the $t$-th task by iteratively moving in the direction of the proposed mixed stochastic gradient.

In the lifelong learning setting, the learning of task $t$ is conducted over a single pass of the training examples in an online fashion. To establish the tradeoff between the performance on old tasks and the $t$-th task, we consider the following optimization problem with composite objective:

$$\min_{\mathbf{w}} \alpha_1 \ell_t(\mathbf{w}) + \alpha_2 \ell_{\text{ref}}(\mathbf{w}) := \mathbb{E}_{\xi,\zeta} \left[ \alpha_1 \ell_t(\mathbf{w}; \xi) + \alpha_2 \ell_{\text{ref}}(\mathbf{w}; \zeta) \right], \tag{4}$$

where $\mathbf{w} \in \mathbb{R}^d$ is the parameter of the model, $\xi, \zeta$ are random variables with finite support, $\ell_t(\mathbf{w})$ is the expected training loss of the $t$-th task, $\ell_{\text{ref}}(\mathbf{w})$ is the expected loss calculated on the data stored in the episodic memory, $\alpha_1$ and $\alpha_2$ are hyperparameters which control the relative importance of $\ell_t(\mathbf{w})$ and $\ell_{\text{ref}}(\mathbf{w})$. Intuitively, a larger $\ell_{\text{ref}}(\mathbf{w})$ signifies catastrophic forgetting. Note that during the learning process of each task, every data sample is i.i.d., and hence the current example of task $t$ could be viewed as a random sample due to online-to-batch conversion argument (Cesa-Bianchi et al., 2004). In the lifelong learning setting, $\ell_t(\mathbf{w}; \xi)$ is the training loss calculated on the current mini-batch (controlled by $\xi$) of task $t$, $\ell_{\text{ref}}(\mathbf{w}; \zeta)$ is the loss calculated on a random mini-batch (controlled by $\zeta$) sampled from the episodic memory. In the traditional online learning, the loss is calculated only based on the current received example but not on historical samples, and hence $\alpha_1 = 1, \alpha_2 = 0$. In this case, the weights are only optimized for the current task while ignoring previous tasks which leads to catastrophic forgetting. If $\alpha_2 > 0$, this formulation naturally involves examples from old tasks. In lifelong learning, since we need to consider performance on both old tasks and the current task, we typically do not consider the degenerate case when $\alpha_2 = 0$.

We use $\mathbf{w}_k^t$ to denote the weight when the model is being trained on the $k$-th mini-batch of task $t$. Clearly, both $\ell_t(\mathbf{w})$ and $\ell_{\text{ref}}(\mathbf{w})$ are determined by $\mathbf{w}_k^t$ during training. This implies that the relative value of $\ell_t(\mathbf{w})$ and $\ell_{\text{ref}}(\mathbf{w})$ is changing between mini-batches. Therefore, $\alpha_1$ and $\alpha_2$ should be adjusted adaptively based on $\mathbf{w}_k^t$ in order to achieve a good balance between old tasks and the current task. To this end, with a little abuse of notation, we define two parameter-dependent functions $\alpha_1, \alpha_2 : \mathbb{R}^d \mapsto \mathbb{R}_+$ to characterize the relative importance of the current task and old tasks. Mathematically, we propose to use the following update:

$$\mathbf{w}_{k+1}^t = \arg\min_{\mathbf{w}} \alpha_1(\mathbf{w}_k^t) \cdot \ell_t(\mathbf{w}) + \alpha_2(\mathbf{w}_k^t) \cdot \ell_{\text{ref}}(\mathbf{w}), \tag{5}$$

**Algorithm 1** MEGA, the proposed algorithm for lifelong learning. $T$ is the number of tasks. $n_t$ is the number of mini-batches of task $t$. $M$ is the episodic memory. $\xi_k^t$ is the $k$-th mini-batch of task $t$ and $y_k^t$ is the corresponding label. $\zeta_k^t$ is a random mini-batch from the episodic memory. $\mathbf{w}_k^t$ stands for the parameter after $k$-th mini-batch during the training of $t$-th task. $\ell_t(\mathbf{w}_k^t; \xi_k^t)$ is the training loss calculated on $\xi_k^t$. $\ell_{\text{ref}}(\mathbf{w}_k^t; \zeta_k^t)$ is the reference loss calculated on $\zeta_k^t$. $\alpha_1$ and $\alpha_2$ are defined in Eq. 5

```
 1: M ← {}
 2: for t ← 1 to T do
 3:     for k ← 1 to n_t do
 4:         if M ≠ {} then
 5:             ζ_k^t ← SAMPLE(M)
 6:             θ̃ ← arccos(∇ℓ_t(w_k^t; ξ_k^t), ∇ℓ_ref(w_k^t; ζ_k^t))
 7:             Solve the optimization problem in Eq. (9) to obtain θ.
 8:             Obtain α_1 and α_2 as in Appendix A.1.
 9:         else
10:             Set α_1 = 1 and α_2 = 0.
11:         end if
12:         Update w_k^t using Eq. 6.
13:         M ← M ⋃(ξ_k^t, y_k^t)
14:         Discard the samples added initially if M is full.
15:     end for
16: end for
```

where $\alpha_1, \alpha_2 : \mathbb{R}^d \mapsto \mathbb{R}_+$ are real-valued mappings.

We employ first-order methods (e.g., stochastic gradient descent) to approximately solve the optimization problem (5 which naturally motivates us to design the MixEd stochastic GrAdient (MEGA) algorithm. MEGA is doing update (6) to approximately solve (5), where one-step stochastic gradient descent is performed with the initial point set to be $\mathbf{w}_k^t$:

$$\mathbf{w}_{k+1}^t \leftarrow \mathbf{w}_k^t - \eta \left( \alpha_1(\mathbf{w}_k^t) \nabla \ell_t(\mathbf{w}_k^t; \xi_k^t) + \alpha_2(\mathbf{w}_k^t) \nabla \ell_{\text{ref}}(\mathbf{w}_k^t; \zeta_k^t) \right), \tag{6}$$

where $\eta$ is the learning rate, $\xi_k^t$ and $\zeta_k^t$ are random variables with finite support, $\nabla \ell_t(\mathbf{w}_k^t; \xi_k^t)$ and $\nabla \ell_{\text{ref}}(\mathbf{w}_k^t; \zeta_k^t)$ are unbiased estimators of $\nabla \ell_t(\mathbf{w}_k^t)$ and $\nabla \ell_{\text{ref}}(\mathbf{w}_k^t)$ respectively, and $\alpha_1(\mathbf{w}_k^t) \nabla \ell_t(\mathbf{w}_k^t; \xi_k^t) + \alpha_2(\mathbf{w}_k^t) \nabla \ell_{\text{ref}}(\mathbf{w}_k^t; \zeta_k^t)$ is referred to as the *mixed stochastic gradient*.

The main difficulty of the update (6) is to define well-behaved mappings $\alpha_1(\cdot)$ and $\alpha_2(\cdot)$ which are consistent with the goal of lifelong learning. To this end, we introduce two approaches, angle-based approach (Section 4.1) and a direct approach (Section 4.2) to address this difficulty. It is worth mentioning that several recent advances on lifelong learning (Lopez-Paz et al., 2017; Chaudhry et al., 2018b; Riemer et al., 2018) have close relationship with the angle-based approach in our MEGA framework which will be illustrated in Section 4.1.

## 4.1 Angle-Based Approach

Note that the mixed stochastic gradient is a linear combination of $\nabla \ell_t(\mathbf{w}_k^t; \xi_k^t)$ and $\nabla \ell_{\text{ref}}(\mathbf{w}_k^t; \zeta_k^t)$. While keeping the magnitude the same as $\nabla \ell_t(\mathbf{w}_k^t; \xi_k^t)$, geometrically the mixed stochastic gradient can be viewed as an appropriate rotation of $\nabla \ell_t(\mathbf{w}_k^t; \xi_k^t)$ by a desired angle $\theta$. This perspective leads to the angle-based approach in our framework. The key idea of the angle-based approach is to first appropriately rotate the stochastic gradient calculated on the current task (i.e., $\nabla \ell_t(\mathbf{w}_k^t; \xi_k^t)$) by an angle $\theta_k^t$, and then use the rotated vector as the mixed stochastic gradient to conduct the update (6) in each mini-batch. For simplicity, we omit the subscript $k$ and superscript $t$ later on unless specified.

We use $\mathbf{g}_{\text{mix}}$ to denote the desired mixed stochastic gradient which has the same magnitude as $\nabla \ell_t(\mathbf{w}; \xi)$. Specifically, we look for the mixed stochastic gradient $\mathbf{g}_{\text{mix}}$ which direction aligns well with both $\nabla \ell_t(\mathbf{w}; \xi)$ and $\nabla \ell_{\text{ref}}(\mathbf{w}; \zeta)$. Mathematically, we want to maximize

$$\frac{\langle \mathbf{g}_{\text{mix}}, \nabla \ell_t(\mathbf{w}; \xi) \rangle}{\|\mathbf{g}_{\text{mix}}\|_2 \cdot \|\nabla \ell_t(\mathbf{w}; \xi)\|_2} + \frac{\langle \mathbf{g}_{\text{mix}}, \nabla \ell_{\text{ref}}(\mathbf{w}; \zeta) \rangle}{\|\mathbf{g}_{\text{mix}}\|_2 \cdot \|\nabla \ell_{\text{ref}}(\mathbf{w}; \zeta)\|_2}, \tag{7}$$

which is equivalent to find an angle $\theta$ such that

$$\theta \in \arg\max_{\beta \in [0, \frac{\pi}{2}]} \cos(\beta) + \cos(\tilde{\theta} - \beta). \tag{8}$$

where $\tilde{\theta} \in [0, \pi]$ is the angle between $\nabla\ell_t(\mathbf{w}; \xi)$ and $\nabla\ell_{\text{ref}}(\mathbf{w}; \zeta)$. To capture the relative importance of the current task and old tasks which is crucial for lifelong learning, we introduce $\ell_t(\mathbf{w}; \xi)$ and $\ell_{\text{ref}}(\mathbf{w}; \zeta)$ into (8),

$$\theta \in \arg\max_{\beta \in [0, \frac{\pi}{2}]} \ell_t(\mathbf{w}; \xi) \cos(\beta) + \ell_{\text{ref}}(\mathbf{w}; \zeta) \cos(\tilde{\theta} - \beta). \tag{9}$$

Here we give some discussions of several special cases of Eq. (9),

- When $\ell_{\text{ref}}(\mathbf{w}; \zeta) = 0$, then $\theta = 0$, and in this case $\alpha_1(\mathbf{w}) = 1$, $\alpha_2(\mathbf{w}) = 0$ in (6), which means the mixed stochastic gradient reduces to $\nabla\ell_t(\mathbf{w}; \xi)$. In the lifelong learning setting, $\ell_{\text{ref}}(\mathbf{w}; \zeta) = 0$ implies that there is almost no catastrophic forgetting, and hence we can update the model parameters exclusively for the current task by moving in the direction of $\nabla\ell_t(\mathbf{w}; \xi)$.

- When $\ell_t(\mathbf{w}; \xi) = 0$, then $\theta = \tilde{\theta}$, and in this case $\alpha_1(\mathbf{w}) = 0$, $\alpha_2(\mathbf{w}) = \|\nabla\ell_t(\mathbf{w}; \xi)\|_2 / \|\nabla\ell_{\text{ref}}(\mathbf{w}; \zeta)\|_2$, provided that $\|\nabla\ell_{\text{ref}}(\mathbf{w}; \zeta)\|_2 \neq 0$ (define 0/0=0). This means the direction of the mixed stochastic gradient is the same as the stochastic gradient calculated on the data in the episodic memory (i.e., $\ell_{\text{ref}}(\mathbf{w}; \zeta)$). In the lifelong learning setting, this update can help improve the performance on old tasks, i.e., avoid catastrophic forgetting.

In the general case, we assume $\ell_{\text{ref}}(\mathbf{w}; \zeta)$ and $\ell_t(\mathbf{w}; \xi)$ are both positive (the edge cases are covered in the above discussion). Since the optimization problem (9) is possibly nonconvex, we propose to use projected gradient ascent to approximately solve it. Mathematically, we do multiple updates using the following formula,

$$\beta \leftarrow \Pi_{[0, \frac{\pi}{2}]} \left[ \beta + \frac{1}{1+r} g(\beta) \right]. \tag{10}$$

where $\Pi$ is the projection operator, $g(\beta) = -\sin(\beta) + \frac{\ell_{\text{ref}}(\mathbf{w}; \zeta)}{\ell_t(\mathbf{w}; \xi)} \sin(\tilde{\theta} - \beta)$, $r = \frac{\ell_{\text{ref}}(\mathbf{w}; \zeta)}{\ell_t(\mathbf{w}; \xi)}$. It is not difficult to show that the smoothness parameter of the function $g'(\beta) = \cos(\beta) + \frac{\ell_{\text{ref}}(\mathbf{w}; \zeta)}{\ell_t(\mathbf{w}; \xi)} \cos(\tilde{\theta} - \beta)$ is $1 + r$, and hence projected gradient ascent can converge to a stationary point (Nesterov, 1998). To avoid getting stuck on saddle point or local minima or local maxima, we can use multiple random starting points and select the one which achieves the largest function value. This strategy is proven to be successful in our experiments.

After we find the desired angle $\theta$, it is easy to obtain $\alpha_1(\mathbf{w})$ and $\alpha_2(\mathbf{w})$ in Eq. (6). For details, please refer to Appendix A.1. Please see Algorithm 1 for the summary of the algorithm.

**Comparison with Existing Works**    There are several existing works setting $\theta$ in different manners. In Lopez-Paz et al. (2017) and Chaudhry et al. (2018b), $\theta = \tilde{\theta}$ if $\tilde{\theta} \leq \frac{\pi}{2}$, and $\theta = \tilde{\theta} - \frac{\pi}{2}$ if $\tilde{\theta} \geq \frac{\pi}{2}$. Note that $\nabla\ell_{\text{ref}}(\mathbf{w}; \zeta)$ is defined differently in Lopez-Paz et al. (2017) and Chaudhry et al. (2018a). In Lopez-Paz et al. (2017), $\nabla\ell_{\text{ref}}(\mathbf{w}; \zeta)$ is calculated on the data of each task separately stored in the episodic memory. While in Chaudhry et al. (2018a), $\nabla\ell_{\text{ref}}(\mathbf{w}; \zeta)$ is computed on a random mini-batch sampled from the episodic memory.

Our work differs from Lopez-Paz et al. (2017); Chaudhry et al. (2018b) in two aspects. First, we explicitly maximize the correlation between the mixed stochastic gradient $\mathbf{g}_{\text{mix}}$ and the current gradient $\nabla\ell_t(\mathbf{w}; \xi)$ (calculated on current data), and the correlation between the mixed stochastic gradient $\mathbf{g}_{\text{mix}}$ and the reference gradient $\nabla\ell_{\text{ref}}(\mathbf{w}; \zeta)$ (calculated on memory data) as in Eq (7). Intuitively, the direction of $\mathbf{g}_{\text{mix}}$ will not bias towards the current task or old tasks (since it is easy to see that the angle between $\mathbf{g}_{\text{mix}}$ and $\nabla\ell_t(\mathbf{w}; \xi)$, and the angle between $\mathbf{g}_{\text{mix}}$ and $\nabla\ell_{\text{ref}}(\mathbf{w}; \zeta)$ are both acute (except the edge case where the angle between $\nabla\ell_t(\mathbf{w}; \xi)$ and $\nabla\ell_{\text{ref}}(\mathbf{w}; \zeta)$ is 180 degree). By following $\mathbf{g}_{\text{mix}}$, each update tends to degrade the loss on both current task and old tasks. While in previous works (Lopez-Paz et al., 2017; Chaudhry et al., 2018b), the corresponding $\mathbf{g}_{\text{mix}}$ is found by projecting the current gradient to be perpendicular to $\nabla\ell_{\text{ref}}(\mathbf{w}; \zeta)$ when the angle between them is obtuse

(in this case, $\mathbf{g}_{\text{mix}} \cdot \nabla \ell_{\text{ref}}(\mathbf{w}; \zeta) = 0$). Intuitively, both GEM and A-GEM put more emphasis on the current task (when $\mathbf{g}_{\text{mix}}$ is perpendicular to $\nabla \ell_{\text{ref}}(\mathbf{w}; \zeta)$, the update following the direction of $\mathbf{g}_{\text{mix}}$ will not directly reduce the loss on old tasks). Second, we introduce the loss on the current batch and on the sampled batch from episodic memory. This allows us to better balance the performance on the current task and old tasks. Since it is preferable to put more emphasis on the tasks (task) which achieve(s) low performance.

## 4.2 DIRECT APPROACH

We also introduce MEGA-D which is a direct approach for implementing MEGA. In MEGA-D, instead of rotating the stochastic gradient computed on the data in the current mini-batch as in the angle-based approach, we define $\alpha_1(\cdot)$ and $\alpha_2(\cdot)$ in the definition of mixed stochastic gradient in a direct manner. Specifically, in the update of (6), we set $\alpha_1 = 1$, $\alpha_2 = \ell_{\text{ref}}(\mathbf{w}; \zeta)/\ell_t(\mathbf{w}; \xi)$ if $\ell_t(\mathbf{w}; \xi) > 0$, and $\alpha_1 = 0$, $\alpha_2 = 1$ if $\ell_t(\mathbf{w}; \xi) = 0$. Intuitively, if $\ell_t(\mathbf{w}; \xi) = 0$, then it means that the model performs well on the current task and can focus on improving performance on the data stored in the episodic memory, and hence $\alpha_1 = 0$, $\alpha_2 = 1$. Otherwise, we keep the balance of the two terms of mixed stochastic gradient according to $\ell_t(\mathbf{w}; \xi)$ and $\ell_{\text{ref}}(\mathbf{w}; \zeta)$. MEGA-D is a simple variant of MEGA which balances current task and old tasks by only leveraging loss information.

## 5 EXPERIMENTS

### 5.1 DATASETS

In the experiments, we consider the following four conventional lifelong learning benchmarks,

- **Permuted MNIST** (Kirkpatrick et al., 2017): this is a variant of standard MNIST dataset (LeCun et al., 1998) of handwritten digits with 20 tasks. Each task has a fixed random permutation of the input pixels which is applied to all the images of that task.
- **Split CIFAR** (Zenke et al., 2017): this dataset consists of 20 disjoint subsets of CIFAR-100 dataset (Krizhevsky et al., 2009), where each subset is formed by randomly sampling 5 classes without replacement from the original 100 classes.
- **Split CUB** (Chaudhry et al., 2018b): the CUB dataset (Wah et al., 2011) is split into 20 disjoint subsets by randomly sampling 10 classes without replacement from the original 200 classes.
- **Split AWA** (Chaudhry et al., 2018b): this dataset consists of 20 subsets of the AWA dataset (Lampert et al., 2009). Each subset is constructed by sampling 5 classes with replacement from a total of 50 classes. Note that the same class can appear in different subsets. As in Chaudhry et al. (2018b), in order to guarantee that each training example only appears once in the learning process, based on the occurrences in different subsets the training data of each class is split into disjoint sets.

We also include **Many Permutations** which is a variant of **Permuted MNIST** to introduce more non-stationality into the learning process. In **Many Permutations**, there are a total 100 tasks with 200 examples per task. The way to generate the tasks is the same as in **Permuted MNIST**, that is, a fixed random permutation of input pixels is applied to all the examples for a particular task.

### 5.2 NETWORK ARCHITECTURES

To be consistent with the previous works (Lopez-Paz et al., 2017; Chaudhry et al., 2018b), for **Permuted MNIST** we adopt a standard fully-connected network with two hidden layers. Each layer has 256 units with ReLU activation. For **Split CIFAR** we use a reduced ResNet18. For **Split CUB** and **Split AWA**, we use a standard ResNet18 (He et al., 2016).

### 5.3 BASELINES AND EXPERIMENTAL SETTINGS

We compare the proposed MEGA with several state-of-the-art lifelong learning methods,

- VAN: in VAN, a single network is trained continuously on a sequence of tasks in a standard supervised learning manner.

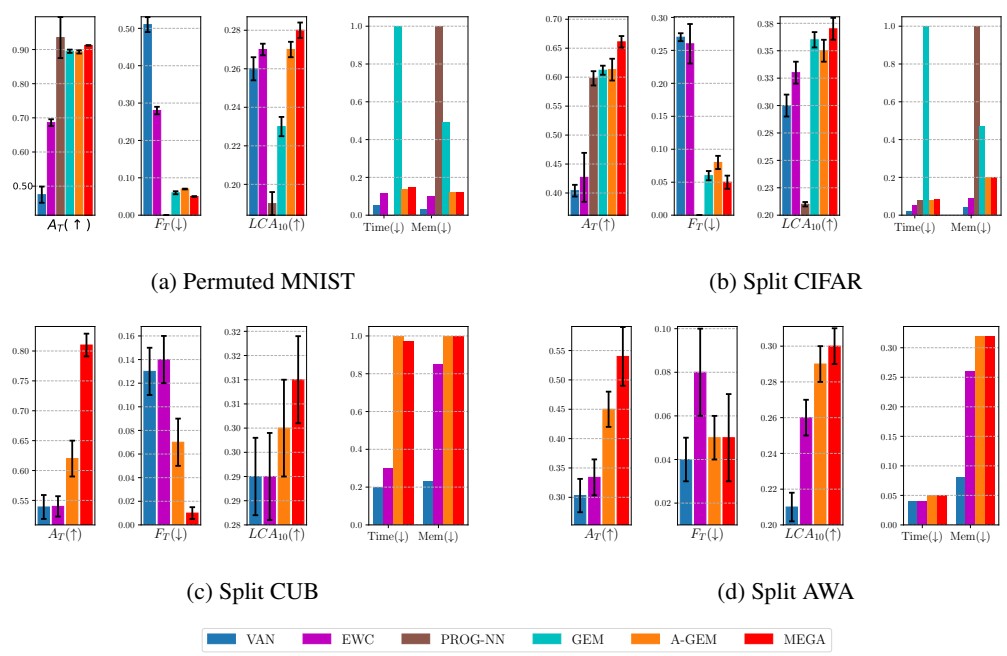

(a) Permuted MNIST

(b) Split CIFAR

(c) Split CUB

(d) Split AWA

VAN      EWC      PROG-NN      GEM      A-GEM      MEGA

Figure 1: Performance of lifelong learning models across different measures on **Permuted MNIST**, **Split CIFAR**, **Split CUB** and **Split AWA**.

- MULTI-TASK: in MULTI-TASK, a single network is trained on the shuffled data from all the tasks with a single pass.

- Episodic memory based approach: GEM (Lopez-Paz et al., 2017) and AGEM (Chaudhry et al., 2018b) are episodic memory based approaches which modify the current gradient when its angle between the gradient computed on the episodic memory is obtuse. MER (Riemer et al., 2018) is another recently proposed episodic memory based approach which maintains an experience replay style memory with reservoir sampling and employs a meta-learning style training strategy.

- Regularization-based approaches: EWC (Kirkpatrick et al., 2017), PI (Zenke et al., 2017), RWALK (Chaudhry et al., 2018a) and MAS (Aljundi et al., 2018) are regularization-based approaches which prevent the important weights of the old tasks from changing too much.

- Knowledge transfer based approach: in PROG-NN (Rusu et al., 2016), a new column with lateral connections with previous hidden layers is added for each new task. This allows knowledge transfer between old tasks and the new task.

To be consistent with Chaudhry et al. (2018b), for episodic memory based approaches, the episodic memory size for each task is 250, 65, 50, and 100, and the batch size for computing the gradients on the episodic memory (if needed) is 256, 1300, 128 and 128 for MNIST, CIFAR, CUB and AWA, respectively. To fill the episodic memory, the examples are chosen uniformly at random for each task as in Chaudhry et al. (2018b). For each dataset, 17 tasks are used for training and 3 tasks are used for hyperparameter search. For MEGA and MER Riemer et al. (2018), we do not conduct exhaustive hyperparameter search and reuse the hyperparameters of AGEM (Chaudhry et al., 2018b). For other baselines, we use best hyperparameters found by Chaudhry et al. (2018b). For the detailed hyperparameters, please see Appendix G of Chaudhry et al. (2018b). In MEGA, we solve Eq.9 three times with different random starting points and the update in Eq.10 is repeated for ten iterations.

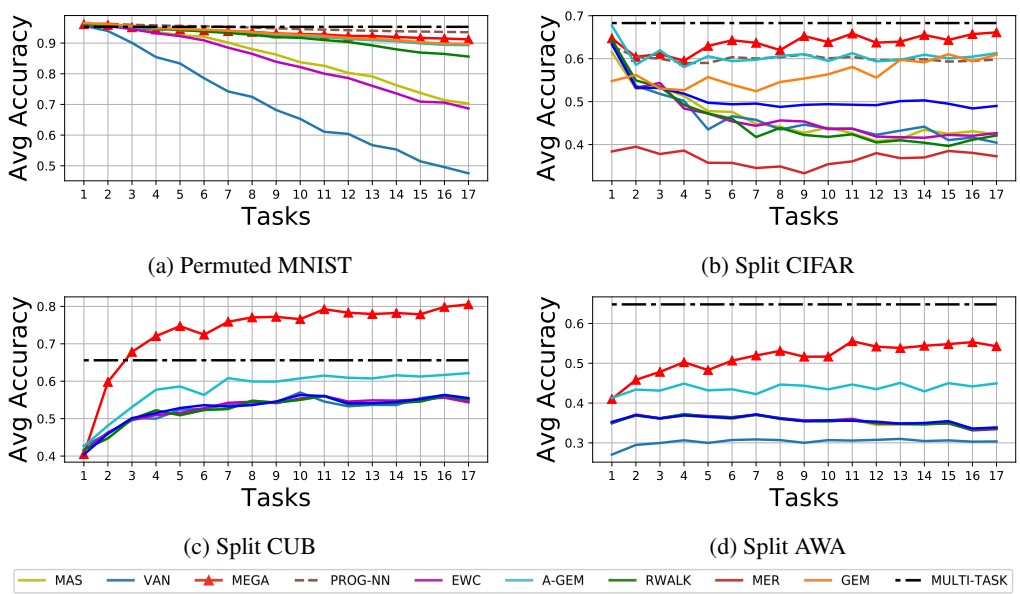

Figure 2: Evolution of average accuracy during the lifelong learning process.

# 6 RESULTS

## 6.1 MEGA VS BASELINES

In Fig. 1 we show the results across different measures on all the benchmark datasets. We have the following observations based on the results. First, the proposed MEGA outperforms all baselines across the benchmarks, except that PROG-NN achieves a slightly higher accuracy on **Permuted MNIST**. As we can see from the memory comparison, PROG-NN is very memory inefficient since it allocates a new network for each task, thus the number of parameters grows super-linearly with the number of tasks. This becomes problematic when large networks are being used. For example, PROG-NN runs out of memory on **Split CUB** and **Split AWA** which prevents it from scaling up to real-life problems. On other datasets, MEGA consistently performs better than all the baselines, from Fig. 2 we can see that on **Split CUB**, MEGA even surpasses the multi-task baseline which is previously believed as an upper bound performance of lifelong learning algorithms (Chaudhry et al., 2018b). Second, MEGA achieves the lowest Forgetting Measure across all the datasets which indicates its ability to overcome catastrophic forgetting. Third, the proposed MEGA also obtains a high LCA across all the datasets which shows that MEGA also learns quickly. The evolution of LCA in the first ten mini-batches across all the datasets is shown in Fig. 3. Last, compared with AGEM (Chaudhry et al., 2018b), which is the state-of-the-art method for lifelong learning, MEGA has the same memory cost and similar time complexity. For detailed results, please refer to Table 4 and Table 5 in Appendix A.2.

In Fig. 2 we show the evolution of average accuracy during the lifelong learning process. As more tasks are added, while the average accuracy of the baselines generally drops due to catastrophic forgetting, MEGA can maintain and even improve its performance. This shows MEGA has a clear advantage over the state-of-the-art lifelong learning methods.

## 6.2 LONG-TERM REMEMBERING

In Table 1 we show the results of Long-term Remembering (LTR) of some representative lifelong learning methods on different datasets. As stated before, an algorithm with low LTR indicates that it can maintain the performance on the tasks trained initially. From Table 1 we can see that the proposed MEGA algorithm achieves the lowest LTR across all the datasets. This demonstrates that MEGA can learn tasks in succession without forgetting the initial tasks which is crucial for real-world lifelong learning applications.

Table 1: Results of Long-term Remembering (LTR).

| Methods | Permuted MNIST | Split CIFAR | Split CUB | Split AWA |
|---|---|---|---|---|
| MEGA | **0.524** $\pm$ 0.017 | **0.356** $\pm$ 0.114 | **0.002** $\pm$ 0.002 | **0.070** $\pm$ 0.114 |
| AGEM | 0.716 $\pm$ 0.048 | 0.643 $\pm$ 0.124 | 0.456 $\pm$ 0.174 | 0.178 $\pm$ 0.082 |
| EWC | 3.292 $\pm$ 0.135 | 2.493 $\pm$ 0.427 | 1.021 $\pm$ 0.210 | 0.675 $\pm$ 0.214 |
| VAN | 5.375 $\pm$ 0.194 | 2.613 $\pm$ 0.174 | 0.976 $\pm$ 0.215 | 0.202 $\pm$ 0.090 |

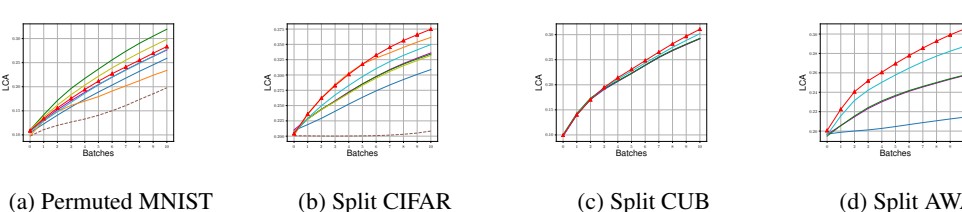

(a) Permuted MNIST      (b) Split CIFAR      (c) Split CUB      (d) Split AWA

Figure 3: LCA of first ten mini-batches on different datasets.

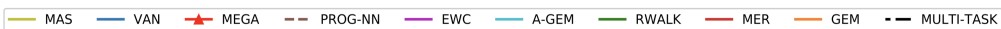

## 6.3 DIRECT APPROACH

We show the comparison of MEGA and MEGA-D in Table 2. The performance of MEGA-D is on par with MEGA across all the datasets. This shows that it is important to explicitly consider the loss on the episodic memory in order to overcome catastrophic forgetting.

Table 2: Comparison of MEGA and MEGA-D.

| Methods | Permuted MNIST | | Split CIFAR | | Split CUB | | Split AWA | |
|---|---|---|---|---|---|---|---|---|
| | $A_T$ (%) | $F_T$ | $A_T$(%) | $F_T$ | $A_T$(%) | $F_T$ | $A_T$(%) | $F_T$ |
| MEGA | 91.21 $\pm$ 0.10 | 0.05 $\pm$ 0.01 | 66.12 $\pm$ 1.93 | 0.05 $\pm$ 0.02 | 80.58 $\pm$ 1.94 | 0.01 $\pm$ 0.01 | 54.28 $\pm$ 4.84 | 0.04 $\pm$ 0.04 |
| MEGA-D | 91.14 $\pm$ 0.16 | 0.05 $\pm$ 0.02 | 66.72 $\pm$ 1.50 | 0.04 $\pm$ 0.01 | 79.68 $\pm$ 2.37 | 0.01 $\pm$ 0.02 | 54.67 $\pm$ 4.69 | 0.04 $\pm$ 0.03 |

## 6.4 MANY PERMUTATION

We show the results on **Many Permutation** in Table 3. Compared with **Permuted MNIST**, **Many Permutation** has 5 times more tasks (100) and much fewer examples per task (200). This introduces more non-stationarity into the learning process. Nevertheless, the proposed MEGA achieves competitive results in this setting. Compared with MER which achieves similar results to MEGA, MEGA is much more time efficient since it does not rely on the meta-learning procedure. Update MEGA-D result in the table.

| Methods | VAN | EWC | GEM | AGEM | MER | MEGA-D | MEGA |
|---|---|---|---|---|---|---|---|
| Average Accuracy (%) | 32.62 $\pm$ 0.43 | 33.46 $\pm$ 0.46 | 56.76 $\pm$ 0.29 | 34.15 $\pm$ 0.55 | **62.52** $\pm$ 0.32 | 56.52$\pm$ 0.43 | **62.48** $\pm$ 0.51 |

Table 3: Results on **Many Permutation**.

## 7 CONCLUSION

In this paper, we present a lifelong learning algorithm called MEGA which achieves the state-of-the-art performance across all the benchmark datasets. In MEGA, we cast the lifelong learning problem as an optimization problem with composite objective and solve it with the proposed mixed stochastic gradient. We also propose an important lifelong learning metric called LTR to characterize the ability of lifelong learning algorithms to maintain performance on the tasks trained in the far past. Extensive experimental results show that the proposed MEGA achieves superior results across all the considered metrics and establishes the new state-of-the-art on all the datasets.

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

# A    APPENDIX

## A.1    SOME DERIVATIONS

For notation simplicity, we use $\mathbf{g}$, $\hat{\mathbf{g}}$, $a$, $b$ to replace $\nabla \ell_t(\mathbf{w}; \xi)$, $\nabla \ell_{\text{ref}}(\mathbf{w}; \zeta)$, $\alpha_1(\mathbf{w})$, $\alpha_2(\mathbf{w})$ respectively. If $\mathbf{g} = \hat{\mathbf{g}}$, then $a = 1$, $b = 0$. Otherwise, the goal is to solve

$$
\begin{aligned}
a\mathbf{g}^\top \mathbf{g} + b\mathbf{g}^\top \hat{\mathbf{g}} &= \|\mathbf{g}\|_2^2 \cos\theta \\
a\mathbf{g}^\top \hat{\mathbf{g}} + b\|\hat{\mathbf{g}}\|_2^2 &= \|\mathbf{g}\|\|\hat{\mathbf{g}}\| \cos(\tilde{\theta} - \theta)
\end{aligned}
\tag{11}
$$

The solution of (11) is

$$
\begin{aligned}
a &= \frac{1}{\|\mathbf{g}\|_2^2 \|\hat{\mathbf{g}}\|_2^2 - \mathbf{g}^\top \hat{\mathbf{g}}} \left[ \|\hat{\mathbf{g}}\|_2^2 \|\mathbf{g}\|_2^2 \cos\theta - (\mathbf{g}^\top \hat{\mathbf{g}})\|\mathbf{g}\|_2 \|\hat{\mathbf{g}}\|_2 \cos(\tilde{\theta} - \theta) \right] \\
b &= \frac{1}{\|\mathbf{g}\|_2^2 \|\hat{\mathbf{g}}\|_2^2 - \mathbf{g}^\top \hat{\mathbf{g}}} \left[ -(\mathbf{g}^\top \hat{\mathbf{g}})\|\mathbf{g}\|_2^2 \cos\theta + \|\mathbf{g}\|_2^3 \|\hat{\mathbf{g}}\|_2 \cos(\tilde{\theta} - \theta) \right]
\end{aligned}
\tag{12}
$$

## A.2    RESULT TABLES

Table 4: The results of Average Accuracy ($A_T$), Forgetting Measure ($F_T$) and LCA of different methods on **Permuted MNIST** and **Split CIFAR**. The results are averaged across 5 runs with different random seeds.

| Methods | Permuted MNIST | | | Split CIFAR | | |
|---|---|---|---|---|---|---|
| | $A_T(\%)$ | $F_T$ | $LCA_{10}$ | $A_T(\%)$ | $F_T$ | $LCA_{10}$ |
| VAN | 47.55±2.37 | 0.52±0.026 | 0.259±0.005 | 40.44±1.02 | 0.27±0.006 | 0.309±0.011 |
| EWC | 68.68±0.98 | 0.28±0.010 | 0.276±0.002 | 42.67±4.24 | 0.26±0.039 | 0.336±0.010 |
| MAS | 70.30±1.67 | 0.26±0.018 | 0.298±0.006 | 42.35±3.52 | 0.26±0.030 | 0.332±0.010 |
| RWALK | 85.60±0.71 | 0.08±0.007 | **0.319**±0.003 | 42.11±3.69 | 0.27±0.032 | 0.334±0.012 |
| MER | - | - | - | 37.27±1.68 | 0.03±0.030 | 0.051±0.101 |
| PROG-NN | **93.55**±0.06 | **0.0**±0.000 | 0.198±0.006 | 59.79±1.23 | **0.0**±0.000 | 0.208±0.002 |
| GEM | 89.50±0.48 | 0.06±0.004 | 0.230±0.005 | 61.20±0.78 | 0.06±0.007 | 0.360±0.007 |
| AGEM | 89.32±0.46 | 0.07±0.004 | 0.277±0.008 | 61.28±1.88 | 0.09±0.018 | 0.350 ±0.013 |
| MEGA | 91.21±0.10 | 0.05±0.001 | 0.283±0.004 | **66.12**±1.94 | 0.06±0.015 | **0.375**±0.012 |

Table 5: The results of Average Accuracy ($A_T$), Forgetting Measure ($F_T$) and LCA of different methods on **Split CUB** and **Split AWA**. The results are averaged across 10 runs with different random seeds.

| Methods | Split CUB | | | Split AWA | | |
|---|---|---|---|---|---|---|
| | $A_T(\%)$ | $F_T$ | $LCA_{10}$ | $A_T(\%)$ | $F_T$ | $LCA_{10}$ |
| VAN | 53.89±2.00 | 0.13±0.020 | 0.292±0.008 | 30.35±2.81 | **0.04**±0.013 | 0.214±0.008 |
| EWC | 53.56±1.67 | 0.14±0.024 | 0.292±0.009 | 33.43±3.07 | 0.08±0.021 | 0.257±0.011 |
| MAS | 54.12±1.72 | 0.13±0.013 | 0.293±0.008 | 33.83±2.99 | 0.08±0.022 | 0.257±0.011 |
| RWALk | 54.11±1.71 | 0.13±0.013 | 0.293±0.009 | 33.63±2.64 | 0.08±0.023 | 0.258±0.011 |
| PI | 55.04±3.05 | 0.12±0.026 | 0.292±0.010 | 33.86±2.77 | 0.08±0.022 | 0.259±0.011 |
| AGEM | 61.82±3.72 | 0.08±0.021 | 0.302±0.011 | 44.95±2.97 | 0.05±0.014 | 0.287±0.012 |
| MEGA | **80.58**±1.94 | **0.01**±0.017 | **0.311**±0.010 | **54.28**±4.84 | 0.05±0.040 | **0.305**±0.015 |

## A.3    ABLATION STUDIES

In this section, we include ablation studies to why the proposed method can improve upon A-GEM Chaudhry et al. (2018b). We only consider maximizing the correlation between the mixed stochastic gradient $\mathbf{g}_{\text{mix}}$ and the current gradient $\nabla \ell_t(\mathbf{w}; \xi)$ (calculated on current data), and the correlation between the mixed stochastic gradient $\mathbf{g}_{\text{mix}}$ and the reference gradient $\nabla \ell_{\text{ref}}(\mathbf{w}; \zeta)$ (calculated on memory data) as in Eq (7). We call this method MEGA-C. The only difference between MEGA-C and A-GEM is the way of modifying the gradient direction. We compare MEGA-C to MEGA and A-GEM in the Table 6. The experimental settings are the same as in Section 5.3.

Table 6: Comparison of MEGA-C, MEGA and A-GEM

| Methods | Permuted MNIST $A_T(\%)$ | Split CIFAR $A_T(\%)$ | Split CUB $A_T(\%)$ | Split AWA $A_T(\%)$ |
|---|---|---|---|---|
| MEGA | $91.21 \pm 0.10$ | $66.12 \pm 1.93$ | $80.58 \pm 1.94$ | $54.28 \pm 4.84$ |
| MEGA-C | $91.15 \pm 0.12$ | $58.04 \pm 1.89$ | $68.60 \pm 1.98$ | $47.95 \pm 4.54$ |
| A-GEM | $89.32 \pm 0.46$ | $61.28 \pm 1.88$ | $61.82 \pm 3.72$ | $44.95 \pm 2.97$ |

From Table 6 we observe that except on **Split CIFAR**, MEGA-C outperforms A-GEM on all the datasets. This demonstrate the benefits of the proposed approach for modifying gradient direction. By considering the loss information as in MEGA, we further improve the results on all the datasets. This shows that both of the two components (gradient direction modification and loss information) contribute to the improvement of the proposed approach.

## A.4 LEARNING PROCESS

In this section we report the result matrices for MEGA and AGEM (Chaudhry et al., 2018b) on each dataset. The entry $(i, j)$ of the matrix is the test accuracy of the $j$-th task after the model is trained on the $i$-th task.

### A.4.1 PERMUTED MNIST

#### MEGA

```
0.9613 0.1091 0.1229 0.0832 0.1374 0.0708 0.0907 0.1017 0.1165 0.1286 0.0979 0.1182 0.1188 0.0886 0.0968 0.0854 0.0928
0.9535 0.9645 0.0895 0.0997 0.1191 0.0685 0.0803 0.1022 0.1165 0.1472 0.1054 0.1112 0.1264 0.1027 0.0872 0.0979 0.0993
0.9391 0.9556 0.9596 0.0996 0.1020 0.0900 0.0807 0.1083 0.0959 0.1400 0.1001 0.1012 0.1096 0.1085 0.0977 0.0716 0.0768
0.9295 0.9473 0.9527 0.9477 0.1113 0.0725 0.0856 0.1033 0.0884 0.1209 0.0847 0.1149 0.1285 0.0939 0.1193 0.0867 0.0824
0.9206 0.9405 0.9437 0.9569 0.9611 0.0785 0.0884 0.1113 0.0926 0.1189 0.0936 0.1337 0.1544 0.1154 0.1282 0.1010 0.0994
0.9119 0.9347 0.9378 0.9481 0.9547 0.9594 0.0916 0.1200 0.1013 0.1042 0.0908 0.1380 0.1415 0.1199 0.1210 0.0908 0.0847
0.9083 0.9261 0.9348 0.9419 0.9478 0.9556 0.9575 0.1092 0.1040 0.1083 0.0863 0.1202 0.1286 0.1177 0.1250 0.0801 0.0931
0.9100 0.9192 0.9291 0.9332 0.9419 0.9462 0.9527 0.9598 0.1152 0.1132 0.0945 0.1054 0.1248 0.1228 0.1187 0.0945 0.0934
0.9022 0.9133 0.9215 0.9271 0.9344 0.9381 0.9430 0.9476 0.9551 0.1187 0.1162 0.1119 0.1364 0.1249 0.1108 0.1012 0.1059
0.8974 0.9074 0.9147 0.9242 0.9289 0.9348 0.9367 0.9404 0.9519 0.9571 0.1102 0.1032 0.1536 0.1261 0.1122 0.1036 0.1093
0.8957 0.9042 0.9146 0.9193 0.9229 0.9313 0.9321 0.9318 0.9409 0.9545 0.9591 0.1144 0.1368 0.1373 0.1143 0.1092 0.1169
0.8863 0.8981 0.9056 0.9127 0.9148 0.9220 0.9270 0.9249 0.9335 0.9444 0.9528 0.9564 0.1517 0.1103 0.1051 0.1002 0.1390
0.8840 0.8992 0.9054 0.9085 0.9149 0.9179 0.9238 0.9198 0.9304 0.9419 0.9441 0.9523 0.9570 0.1292 0.1083 0.0926 0.1301
0.8808 0.8901 0.8994 0.8986 0.9084 0.9113 0.9168 0.9185 0.9239 0.9360 0.9382 0.9453 0.9522 0.9589 0.1017 0.0941 0.1277
0.8770 0.8850 0.8957 0.8926 0.9012 0.9101 0.909 0 0.9132 0.9188 0.9301 0.9358 0.9368 0.9430 0.9508 0.9521 0.0946 0.1334
0.8752 0.8806 0.8911 0.8854 0.8965 0.9070 0.9062 0.9059 0.9145 0.9265 0.9286 0.9338 0.9374 0.9434 0.9462 0.9601 0.1291
0.8732 0.8765 0.8824 0.8809 0.8945 0.9024 0.9016 0.9007 0.9088 0.9202 0.9228 0.9276 0.9279 0.9376 0.9408 0.9521 0.9556
```

#### AGEM

```
0.9613 0.1091 0.1229 0.0832 0.1374 0.0708 0.0907 0.1017 0.1165 0.1286 0.0979 0.1182 0.1188 0.0886 0.0968 0.0854 0.0928
0.9509 0.9645 0.0956 0.0991 0.1304 0.0696 0.0840 0.1033 0.1219 0.1454 0.1064 0.1133 0.1314 0.1043 0.0883 0.0979 0.0973
0.9410 0.9545 0.9615 0.0995 0.0964 0.0921 0.0710 0.1126 0.1176 0.1402 0.1112 0.1026 0.1185 0.1204 0.1077 0.0779 0.0761
0.9299 0.9450 0.9540 0.9546 0.1046 0.0788 0.0959 0.1033 0.1096 0.1266 0.1015 0.1152 0.1476 0.0885 0.1375 0.0984 0.0831
0.9151 0.9361 0.9425 0.9551 0.9588 0.0803 0.0809 0.1143 0.1063 0.1227 0.1066 0.1253 0.1436 0.1154 0.1131 0.1079 0.0915
0.9068 0.9312 0.9401 0.9450 0.9566 0.9590 0.0892 0.1189 0.1285 0.1086 0.1007 0.1433 0.1279 0.1179 0.1097 0.0892 0.0865
0.9015 0.9228 0.9339 0.9385 0.9473 0.9548 0.9586 0.1063 0.1073 0.1102 0.1048 0.1164 0.1291 0.1284 0.1341 0.0854 0.1024
0.8980 0.9155 0.9248 0.9280 0.9356 0.9403 0.9539 0.9580 0.1015 0.1231 0.1129 0.1125 0.1267 0.1133 0.1220 0.0921 0.0985
0.8952 0.9055 0.9201 0.9182 0.9273 0.9310 0.9447 0.9435 0.9512 0.1098 0.1374 0.1166 0.1264 0.1064 0.1183 0.0986 0.1048
0.8846 0.8996 0.9083 0.9154 0.9189 0.9267 0.9363 0.9339 0.9513 0.9558 0.1243 0.1095 0.1179 0.1137 0.1126 0.0945 0.0979
0.8764 0.8977 0.9011 0.9073 0.9086 0.9167 0.9292 0.9274 0.9386 0.9481 0.9631 0.1116 0.1099 0.1417 0.1100 0.0975 0.1166
0.8710 0.8882 0.8937 0.8922 0.9043 0.9077 0.9151 0.9174 0.9279 0.9324 0.9518 0.9572 0.1346 0.1240 0.0964 0.0930 0.1283
0.8625 0.8822 0.8847 0.8855 0.9013 0.8990 0.9093 0.9088 0.9189 0.9295 0.9411 0.9458 0.9533 0.1309 0.1059 0.0987 0.1139
0.8581 0.8784 0.8774 0.8817 0.8954 0.8938 0.8986 0.9003 0.9082 0.9225 0.9307 0.9350 0.9435 0.9603 0.1048 0.1023 0.1048
0.8492 0.8674 0.8732 0.8697 0.8828 0.8826 0.8930 0.8898 0.8962 0.9098 0.9184 0.9267 0.9290 0.9425 0.9542 0.1012 0.1070
```

0.8322 0.8700 0.8644 0.8493 0.8765 0.8787 0.8904 0.8848 0.8883 0.8979 0.9110 0.9158 0.9177 0.9299 0.9403 0.9609 0.1179
0.8438 0.8603 0.8555 0.8488 0.8864 0.8785 0.8798 0.8702 0.8916 0.8968 0.9076 0.9094 0.9092 0.9228 0.9228 0.9463 0.9551

## A.4.2  SPLIT CIFAR

### MEGA

0.6472 0.0000 0.0000 0.0000 0.0000 0.0000 0.0000 0.0000 0.0000 0.0000 0.0000 0.0000 0.0000 0.0000 0.0000 0.0000 0.0000
0.6260 0.5824 0.0000 0.0000 0.0000 0.0000 0.0000 0.0000 0.0000 0.0000 0.0000 0.0000 0.0000 0.0000 0.0000 0.0000 0.0000
0.6324 0.5700 0.6300 0.0000 0.0000 0.0000 0.0000 0.0000 0.0000 0.0000 0.0000 0.0000 0.0000 0.0000 0.0000 0.0000 0.0000
0.6236 0.5496 0.5624 0.6452 0.0000 0.0000 0.0000 0.0000 0.0000 0.0000 0.0000 0.0000 0.0000 0.0000 0.0000 0.0000 0.0000
0.6132 0.5612 0.6048 0.6736 0.6960 0.0000 0.0000 0.0000 0.0000 0.0000 0.0000 0.0000 0.0000 0.0000 0.0000 0.0000 0.0000
0.6140 0.5628 0.5692 0.6632 0.6792 0.7688 0.0000 0.0000 0.0000 0.0000 0.0000 0.0000 0.0000 0.0000 0.0000 0.0000 0.0000
0.5780 0.5500 0.5864 0.6364 0.6792 0.7420 0.6868 0.0000 0.0000 0.0000 0.0000 0.0000 0.0000 0.0000 0.0000 0.0000 0.0000
0.5756 0.5308 0.5764 0.6292 0.6500 0.6820 0.6580 0.6580 0.0000 0.0000 0.0000 0.0000 0.0000 0.0000 0.0000 0.0000 0.0000
0.6212 0.5704 0.5876 0.6376 0.6600 0.7056 0.6636 0.6916 0.7376 0.0000 0.0000 0.0000 0.0000 0.0000 0.0000 0.0000 0.0000
0.5992 0.5580 0.5828 0.6212 0.6528 0.6512 0.6496 0.6700 0.7276 0.6732 0.0000 0.0000 0.0000 0.0000 0.0000 0.0000 0.0000
0.6104 0.5552 0.5804 0.6396 0.6700 0.6960 0.6568 0.6752 0.7412 0.6752 0.7432 0.0000 0.0000 0.0000 0.0000 0.0000 0.0000
0.5880 0.5516 0.5816 0.6248 0.6552 0.6568 0.6412 0.6284 0.7044 0.6304 0.7196 0.6660 0.0000 0.0000 0.0000 0.0000 0.0000
0.6020 0.5628 0.5792 0.6164 0.6444 0.6636 0.6356 0.6536 0.7008 0.6132 0.7000 0.6336 0.7108 0.0000 0.0000 0.0000 0.0000
0.6124 0.5692 0.5924 0.6420 0.6516 0.6912 0.6352 0.6492 0.6848 0.6400 0.6872 0.6312 0.7280 0.7596 0.0000 0.0000 0.0000
0.6012 0.5468 0.5908 0.6128 0.6552 0.6852 0.6288 0.6428 0.6704 0.6176 0.6988 0.6268 0.7088 0.7348 0.6324 0.0000 0.0000
0.6244 0.5588 0.5960 0.6432 0.6448 0.6868 0.6388 0.6540 0.6896 0.6056 0.6920 0.6192 0.7124 0.7344 0.6560 0.7604 0.0000
0.6088 0.5896 0.5840 0.6552 0.6716 0.6904 0.6584 0.6372 0.7032 0.6300 0.6900 0.5864 0.6832 0.7140 0.6348 0.7264 0.7780

### AGEM

0.6772 0.0000 0.0000 0.0000 0.0000 0.0000 0.0000 0.0000 0.0000 0.0000 0.0000 0.0000 0.0000 0.0000 0.0000 0.0000 0.0000
0.5948 0.5764 0.0000 0.0000 0.0000 0.0000 0.0000 0.0000 0.0000 0.0000 0.0000 0.0000 0.0000 0.0000 0.0000 0.0000 0.0000
0.6324 0.5828 0.6432 0.0000 0.0000 0.0000 0.0000 0.0000 0.0000 0.0000 0.0000 0.0000 0.0000 0.0000 0.0000 0.0000 0.0000
0.5980 0.5384 0.5396 0.6456 0.0000 0.0000 0.0000 0.0000 0.0000 0.0000 0.0000 0.0000 0.0000 0.0000 0.0000 0.0000 0.0000
0.5864 0.5404 0.5576 0.6436 0.7004 0.0000 0.0000 0.0000 0.0000 0.0000 0.0000 0.0000 0.0000 0.0000 0.0000 0.0000 0.0000
0.5728 0.5392 0.5068 0.5940 0.6344 0.7180 0.0000 0.0000 0.0000 0.0000 0.0000 0.0000 0.0000 0.0000 0.0000 0.0000 0.0000
0.5572 0.5404 0.5308 0.6224 0.6116 0.6520 0.6688 0.0000 0.0000 0.0000 0.0000 0.0000 0.0000 0.0000 0.0000 0.0000 0.0000
0.6064 0.5356 0.5492 0.5872 0.6164 0.6532 0.6296 0.6724 0.0000 0.0000 0.0000 0.0000 0.0000 0.0000 0.0000 0.0000 0.0000
0.6060 0.5472 0.5528 0.6236 0.5920 0.6348 0.6076 0.6348 0.6972 0.0000 0.0000 0.0000 0.0000 0.0000 0.0000 0.0000 0.0000
0.6004 0.5080 0.4960 0.6128 0.5656 0.6356 0.5752 0.6140 0.6580 0.6792 0.0000 0.0000 0.0000 0.0000 0.0000 0.0000 0.0000
0.5992 0.5408 0.5332 0.5964 0.5928 0.6520 0.5928 0.6304 0.6764 0.5916 0.7364 0.0000 0.0000 0.0000 0.0000 0.0000 0.0000
0.5748 0.5020 0.5104 0.5936 0.6016 0.6184 0.5772 0.6164 0.6408 0.5688 0.6776 0.6436 0.0000 0.0000 0.0000 0.0000 0.0000
0.6056 0.5100 0.5200 0.5916 0.6012 0.6056 0.5816 0.6060 0.6236 0.5808 0.6288 0.5768 0.7332 0.0000 0.0000 0.0000 0.0000
0.6184 0.5344 0.5308 0.5888 0.6116 0.6188 0.6012 0.6248 0.6136 0.5836 0.6428 0.5688 0.6524 0.7392 0.0000 0.0000 0.0000
0.6012 0.5220 0.5488 0.6008 0.5828 0.6048 0.5728 0.5884 0.6356 0.5740 0.6476 0.5540 0.6520 0.6804 0.6460 0.0000 0.0000
0.5984 0.5360 0.5520 0.5808 0.5704 0.6184 0.6068 0.6108 0.6452 0.5404 0.6520 0.5256 0.6624 0.6512 0.5864 0.7388 0.0000
0.6232 0.5356 0.5412 0.6104 0.6080 0.6248 0.5944 0.5900 0.6492 0.5872 0.6468 0.5352 0.6336 0.6520 0.5908 0.6444 0.7508

## A.4.3  SPLIT CUB

### MEGA

0.4050 0.0000 0.0000 0.0000 0.0000 0.0000 0.0000 0.0000 0.0000 0.0000 0.0000 0.0000 0.0000 0.0000 0.0000 0.0000 0.0000
0.6876 0.5088 0.0000 0.0000 0.0000 0.0000 0.0000 0.0000 0.0000 0.0000 0.0000 0.0000 0.0000 0.0000 0.0000 0.0000 0.0000
0.6844 0.7244 0.6263 0.0000 0.0000 0.0000 0.0000 0.0000 0.0000 0.0000 0.0000 0.0000 0.0000 0.0000 0.0000 0.0000 0.0000
0.7071 0.7445 0.7753 0.6553 0.0000 0.0000 0.0000 0.0000 0.0000 0.0000 0.0000 0.0000 0.0000 0.0000 0.0000 0.0000 0.0000
0.7294 0.7483 0.7941 0.7787 0.6857 0.0000 0.0000 0.0000 0.0000 0.0000 0.0000 0.0000 0.0000 0.0000 0.0000 0.0000 0.0000
0.7049 0.7266 0.7712 0.7538 0.7528 0.6366 0.0000 0.0000 0.0000 0.0000 0.0000 0.0000 0.0000 0.0000 0.0000 0.0000 0.0000
0.7271 0.7488 0.7877 0.7893 0.7861 0.7905 0.6830 0.0000 0.0000 0.0000 0.0000 0.0000 0.0000 0.0000 0.0000 0.0000 0.0000
0.7316 0.7737 0.7920 0.7867 0.7851 0.7956 0.8179 0.6816 0.0000 0.0000 0.0000 0.0000 0.0000 0.0000 0.0000 0.0000 0.0000
0.7503 0.7525 0.7860 0.7947 0.7838 0.7818 0.8148 0.7912 0.6934 0.0000 0.0000 0.0000 0.0000 0.0000 0.0000 0.0000 0.0000
0.7445 0.7475 0.7819 0.7599 0.7680 0.7834 0.7901 0.7858 0.7883 0.7085 0.0000 0.0000 0.0000 0.0000 0.0000 0.0000 0.0000
0.7506 0.7732 0.8053 0.8129 0.8049 0.8099 0.8275 0.7938 0.8041 0.8146 0.7211 0.0000 0.0000 0.0000 0.0000 0.0000 0.0000
0.7475 0.7670 0.7860 0.7982 0.7818 0.7892 0.8133 0.7881 0.8014 0.8145 0.7958 0.7144 0.0000 0.0000 0.0000 0.0000 0.0000

0.7393 0.7572 0.7907 0.8006 0.7821 0.8043 0.8123 0.7629 0.7905 0.7902 0.8161 0.7622 0.7262 0.0000 0.0000 0.0000 0.0000
0.7492 0.7501 0.8031 0.8098 0.7956 0.8020 0.8053 0.7811 0.7873 0.8081 0.8159 0.7672 0.7981 0.6794 0.0000 0.0000 0.0000
0.7488 0.7661 0.7958 0.7830 0.7845 0.7873 0.8007 0.7759 0.7902 0.8139 0.8142 0.7746 0.7984 0.7651 0.6831 0.0000 0.0000
0.7538 0.7857 0.8116 0.8170 0.8050 0.8052 0.8277 0.8024 0.8070 0.8220 0.8301 0.7767 0.8210 0.7839 0.7777 0.7505 0.0000
0.7728 0.7846 0.8203 0.7995 0.8141 0.8219 0.8201 0.7973 0.8109 0.8366 0.8390 0.7880 0.8300 0.7948 0.8078 0.8112 0.7418

## AGEM

0.4263 0.0000 0.0000 0.0000 0.0000 0.0000 0.0000 0.0000 0.0000 0.0000 0.0000 0.0000 0.0000 0.0000 0.0000 0.0000 0.0000
0.4383 0.5243 0.0000 0.0000 0.0000 0.0000 0.0000 0.0000 0.0000 0.0000 0.0000 0.0000 0.0000 0.0000 0.0000 0.0000 0.0000
0.4642 0.5220 0.6064 0.0000 0.0000 0.0000 0.0000 0.0000 0.0000 0.0000 0.0000 0.0000 0.0000 0.0000 0.0000 0.0000 0.0000
0.4850 0.5420 0.6057 0.6765 0.0000 0.0000 0.0000 0.0000 0.0000 0.0000 0.0000 0.0000 0.0000 0.0000 0.0000 0.0000 0.0000
0.4935 0.5378 0.6328 0.6042 0.6621 0.0000 0.0000 0.0000 0.0000 0.0000 0.0000 0.0000 0.0000 0.0000 0.0000 0.0000 0.0000
0.4678 0.5071 0.6369 0.5585 0.5966 0.6137 0.0000 0.0000 0.0000 0.0000 0.0000 0.0000 0.0000 0.0000 0.0000 0.0000 0.0000
0.4909 0.5630 0.6435 0.6369 0.6140 0.6278 0.6825 0.0000 0.0000 0.0000 0.0000 0.0000 0.0000 0.0000 0.0000 0.0000 0.0000
0.4928 0.5607 0.6157 0.5992 0.6025 0.6037 0.6572 0.6617 0.0000 0.0000 0.0000 0.0000 0.0000 0.0000 0.0000 0.0000 0.0000
0.4850 0.5401 0.6090 0.5830 0.6004 0.5980 0.6384 0.6520 0.6850 0.0000 0.0000 0.0000 0.0000 0.0000 0.0000 0.0000 0.0000
0.4901 0.5542 0.6086 0.5858 0.6080 0.5985 0.6206 0.6533 0.6607 0.6964 0.0000 0.0000 0.0000 0.0000 0.0000 0.0000 0.0000
0.4909 0.5455 0.6465 0.6220 0.6241 0.6186 0.6094 0.6197 0.6181 0.6569 0.7128 0.0000 0.0000 0.0000 0.0000 0.0000 0.0000
0.5075 0.5675 0.6266 0.6126 0.5999 0.6037 0.6073 0.6214 0.6004 0.6424 0.6547 0.6673 0.0000 0.0000 0.0000 0.0000 0.0000
0.4909 0.5588 0.5943 0.5852 0.5898 0.6018 0.6188 0.5974 0.6266 0.6228 0.6792 0.6281 0.7035 0.0000 0.0000 0.0000 0.0000
0.5297 0.5535 0.6220 0.6281 0.6383 0.6037 0.6234 0.5967 0.6277 0.6051 0.6393 0.5950 0.6965 0.6630 0.0000 0.0000 0.0000
0.5104 0.5677 0.6446 0.6198 0.6135 0.6093 0.6100 0.5710 0.5912 0.6314 0.6312 0.5951 0.6924 0.6422 0.6584 0.0000 0.0000
0.5196 0.5527 0.6350 0.6043 0.6384 0.6009 0.6226 0.5971 0.6028 0.6210 0.6507 0.6318 0.6757 0.6003 0.5899 0.7272 0.0000
0.5345 0.5519 0.6395 0.6064 0.6335 0.6000 0.5879 0.5899 0.5869 0.6509 0.6486 0.6332 0.6914 0.6130 0.5983 0.6806 0.7216

## A.4.4 SPLIT AWA

## MEGA

0.4101 0.0000 0.0000 0.0000 0.0000 0.0000 0.0000 0.0000 0.0000 0.0000 0.0000 0.0000 0.0000 0.0000 0.0000 0.0000 0.0000
0.4791 0.4377 0.0000 0.0000 0.0000 0.0000 0.0000 0.0000 0.0000 0.0000 0.0000 0.0000 0.0000 0.0000 0.0000 0.0000 0.0000
0.5061 0.5174 0.4116 0.0000 0.0000 0.0000 0.0000 0.0000 0.0000 0.0000 0.0000 0.0000 0.0000 0.0000 0.0000 0.0000 0.0000
0.5184 0.5219 0.4962 0.4741 0.0000 0.0000 0.0000 0.0000 0.0000 0.0000 0.0000 0.0000 0.0000 0.0000 0.0000 0.0000 0.0000
0.4931 0.5219 0.4828 0.5142 0.4026 0.0000 0.0000 0.0000 0.0000 0.0000 0.0000 0.0000 0.0000 0.0000 0.0000 0.0000 0.0000
0.5122 0.5357 0.4894 0.5293 0.5053 0.4680 0.0000 0.0000 0.0000 0.0000 0.0000 0.0000 0.0000 0.0000 0.0000 0.0000 0.0000
0.5242 0.5226 0.5038 0.5219 0.5215 0.5605 0.4820 0.0000 0.0000 0.0000 0.0000 0.0000 0.0000 0.0000 0.0000 0.0000 0.0000
0.5383 0.5359 0.5216 0.5323 0.5204 0.5551 0.5774 0.4682 0.0000 0.0000 0.0000 0.0000 0.0000 0.0000 0.0000 0.0000 0.0000
0.5291 0.5179 0.4850 0.5367 0.5043 0.5380 0.5565 0.5111 0.4698 0.0000 0.0000 0.0000 0.0000 0.0000 0.0000 0.0000 0.0000
0.5386 0.5106 0.4995 0.5245 0.5150 0.5470 0.5457 0.4862 0.5380 0.4619 0.0000 0.0000 0.0000 0.0000 0.0000 0.0000 0.0000
0.5612 0.5569 0.5260 0.5566 0.5387 0.5765 0.5847 0.5483 0.5444 0.5622 0.5542 0.0000 0.0000 0.0000 0.0000 0.0000 0.0000
0.5593 0.5593 0.5416 0.5611 0.5345 0.5348 0.5539 0.5358 0.5418 0.5363 0.5720 0.4725 0.0000 0.0000 0.0000 0.0000 0.0000
0.5471 0.5522 0.5333 0.5383 0.5159 0.5410 0.5638 0.5234 0.5234 0.5342 0.5614 0.5611 0.5026 0.0000 0.0000 0.0000 0.0000
0.5557 0.5561 0.5391 0.5368 0.5093 0.5442 0.5743 0.5026 0.5484 0.5126 0.5788 0.5548 0.5641 0.5386 0.0000 0.0000 0.0000
0.5240 0.5563 0.5366 0.5465 0.5102 0.5659 0.5688 0.5071 0.5320 0.5183 0.5865 0.5705 0.5295 0.6206 0.5483 0.0000 0.0000
0.5541 0.5575 0.5219 0.5559 0.5310 0.5490 0.5983 0.5121 0.5466 0.5204 0.5929 0.5626 0.5466 0.6064 0.6291 0.4661 0.0000
0.5542 0.5469 0.5093 0.5677 0.5136 0.5471 0.5581 0.4954 0.5326 0.5169 0.5857 0.5771 0.5345 0.5964 0.6081 0.5187 0.4655

## A-GEM

0.4127 0.0000 0.0000 0.0000 0.0000 0.0000 0.0000 0.0000 0.0000 0.0000 0.0000 0.0000 0.0000 0.0000 0.0000 0.0000 0.0000
0.4256 0.4422 0.0000 0.0000 0.0000 0.0000 0.0000 0.0000 0.0000 0.0000 0.0000 0.0000 0.0000 0.0000 0.0000 0.0000 0.0000
0.4436 0.4445 0.4058 0.0000 0.0000 0.0000 0.0000 0.0000 0.0000 0.0000 0.0000 0.0000 0.0000 0.0000 0.0000 0.0000 0.0000
0.4371 0.4784 0.4334 0.4463 0.0000 0.0000 0.0000 0.0000 0.0000 0.0000 0.0000 0.0000 0.0000 0.0000 0.0000 0.0000 0.0000
0.4339 0.4795 0.4236 0.4258 0.3963 0.0000 0.0000 0.0000 0.0000 0.0000 0.0000 0.0000 0.0000 0.0000 0.0000 0.0000 0.0000
0.4226 0.4674 0.4311 0.4505 0.3864 0.4495 0.0000 0.0000 0.0000 0.0000 0.0000 0.0000 0.0000 0.0000 0.0000 0.0000 0.0000
0.4279 0.4462 0.4268 0.4217 0.3854 0.4254 0.4239 0.0000 0.0000 0.0000 0.0000 0.0000 0.0000 0.0000 0.0000 0.0000 0.0000
0.4621 0.4733 0.4421 0.4563 0.4239 0.4356 0.4489 0.4299 0.0000 0.0000 0.0000 0.0000 0.0000 0.0000 0.0000 0.0000 0.0000
0.4672 0.4774 0.4363 0.4402 0.4265 0.4387 0.4503 0.4129 0.4431 0.0000 0.0000 0.0000 0.0000 0.0000 0.0000 0.0000 0.0000
0.4659 0.4417 0.4385 0.4386 0.4188 0.4419 0.4655 0.4075 0.3971 0.4286 0.0000 0.0000 0.0000 0.0000 0.0000 0.0000 0.0000
0.4555 0.4657 0.4495 0.4645 0.4133 0.4374 0.4717 0.4083 0.4289 0.4144 0.5037 0.0000 0.0000 0.0000 0.0000 0.0000 0.0000

0.4425 0.4463 0.4277 0.4532 0.4301 0.4319 0.4865 0.4288 0.4043 0.4021 0.4172 0.4478 0.0000 0.0000 0.0000 0.0000 0.0000
0.4427 0.4582 0.4438 0.4527 0.4345 0.4638 0.4895 0.4327 0.4181 0.4295 0.4745 0.4303 0.4889 0.0000 0.0000 0.0000 0.0000
0.4538 0.4519 0.4066 0.4696 0.4099 0.4459 0.4859 0.4080 0.3931 0.3827 0.4382 0.3854 0.4055 0.4736 0.0000 0.0000 0.0000
0.4476 0.4786 0.4148 0.4879 0.4269 0.4576 0.5014 0.4532 0.4264 0.4137 0.4535 0.4178 0.3955 0.4809 0.4941 0.0000 0.0000
0.4482 0.4635 0.4114 0.4720 0.4231 0.4527 0.5082 0.4076 0.4231 0.4262 0.4436 0.4122 0.3915 0.4809 0.4762 0.4286 0.0000
0.4530 0.4605 0.4294 0.4519 0.4388 0.4649 0.4885 0.4514 0.4395 0.4236 0.4760 0.4495 0.4105 0.4621 0.4730 0.4039 0.4646

