# OpenReview forum: "Learning with Long-term Remembering: Following the Lead of Mixed Stochastic Gradient"
_ICLR.cc/2020/Conference — Reject_

### Official Review · AnonReviewer3 · 2019-10-21
**Official Blind Review #3**

**Rating:** 3

**Review:**

UPDATE:

I thank the authors for proactively engaging with the review process and improving the paper.

After considering the other reviews and discussions with other reviewers, I also share the concern that the simple MEGA-D baseline performs very well, with little additional gain from the full MEGA approach (only on the many permutations case that was introduced in the rebuttal). Unfortunately, it doesn't look like this point has been fully addressed.

I know this is part of the contribution, and I am certainly an advocate of simple techniques that yield strong results. However, as it stands, this baseline is only mentioned briefly, with a single paragraph in the method section and a single paragraph on evaluation. Given the strength of the result, I think a lot more of the paper should be devoted to understanding the merits of this simple method and evaluating how it relates to the proposed angle-based approach.

I am also a bit confused by the new baseline; given that the memory and current batch are both used for the MEGA-C case, I think the explanation of how this differs from the full case could be clearer.

As such, I must regrettably change my score to a 3. I think this paper has potential; and with a bit more analysis and clarity on the above points, could be a good submission. I encourage the authors to address these for a future publication.

==============================================================

This paper describes an approach to perform continual learning by maintaining an episodic memory / coreset of old examples and learning a linear weighting function between the gradients from new and old samples. A very simple direct method is proposed, as well as an angle-based approach. In the latter, projected gradient ascent is used to find an optimal rotation angle for the current data gradient such that the resulting direction also aligns well with the gradients computed from the memory buffer. This appears to generalise previous work (such as GEM and A-GEM), and a new metric of long-term remembering (LTR) is also introduced.

The experiments are comprehensive and compelling.
The paper is clearly written and easy to follow, and I think it could be quite a good contribution to the conference.

I have some questions and concerns that I think should be addressed first:
1) If I understand correctly, Algorithm 1 seems to indicate that every single batch is added to the memory buffer - I assume this is an error, as it is suggested throughout the paper that only a small buffer is used. How is the memory buffer updated?
2) It is unclear how much memory is required for this approach and whether this is consistent with previous approaches. An ablation over memory size would help with this (ideally with comparison to other episodic memory-based approaches); and a discussion on memory use of different methods is needed.
3) With the direct approach, it seems odd to specify a loss threshold of zero to determine that the current task performance is high. What loss is being used? Further, how does the direct approach relate to eg. GEM/A-GEM/MER in terms of weighting between old and new samples?
4) The related work section is quite thin, and there are several other works that could be cited; currently they seem to be focused on just "gradient-similarity based continual learning" with a few other continual learning works.
5) The paper states that progressive networks increase in memory super-linearly, but I don't believe this is the case; it would be linear or sub-linear, given that new tasks would typically benefit from forward transfer and require fewer additional units.

**Experience Assessment:**

I have published one or two papers in this area.

**Review Assessment: Checking Correctness Of Derivations And Theory:**

I assessed the sensibility of the derivations and theory.

**Review Assessment: Checking Correctness Of Experiments:**

I carefully checked the experiments.

**Review Assessment: Thoroughness In Paper Reading:**

I read the paper thoroughly.

---

> ### Author Response · Authors · 2019-11-10
> **Reply to reviewer 3**
>
> We thank the reviewer for the constructive comments.
>
> Q1: How is the memory buffer updated？
> A: For a fair comparison with GEM [1] and AGEM [2], we update the memory buffer in the same way, i.e., the examples are chosen uniformly at random for each task as in [2]. The buffer can be regarded as a queue, when the buffer is full, old samples are discarded. The current notation is a little misleading, and we will correct it in the updated draft.
>
>
> Q2: Not clear how much memory is required for this approach and whether it is consistent with previous approaches.
> A: Thanks for the suggestions. In the current work, for all the methods we use the same memory size for a fair comparison which is also the setting used in [2]. It is definitely interesting to vary the memory size to see the change of performance of different methods. Below we add results of changing memory size on Permuted MNIST (17 tasks and 60000 examples per task), and compare our approach (MEGA) to A-GEM [2]. We will update the results in the final draft and also conduct similar experiments on other datasets.
>
> Memory size:        850  	1700		2550	 	4250
>
> MEGA:		       86.13     	88.56   	        89.62 		91.21
>
> A-GEM 	               85.60	88.19		88.97		89.32
>
>
> We can see that the proposed MEGA consistently outperform A-GEM regardless of the memory size.
>
> Q3: What loss is used in the direct approach? How does the direct approach relate to eg. GEM/A-GEM/MER in terms of weighting between old and new samples?
> A:  We use the loss on the current batch and a sampled batch from the memory to achieve a better balancing than previous works. Intuitively, our approach put more emphasis on the tasks which achieve a lower performance. The direct approach (MEGA-D) directly leverages the losses to balance the current gradient and the gradient computed on the episodic memory. However, in GEM/A-GEM/MER, the modification of the direction of the current gradient only based on the angle between the gradient and the reference gradient which does not consider the loss information. The connection between GEM/A-GEM/MER, MEGA and MEGA-D can be clearly explained as below,
>
> GEM/A-GEM/MER $\rightarrow^1$ MEGA $\rightarrow^2$ MEGA-D
>
> 1. Better modification of the current gradient direction by maximizing correlation as in Eq 7 and better balancing with losses on current task and old tasks.
>
> 2. Simple variant of MEGA which balances current task and old tasks by only leveraging loss information.
>
>
> Q4: Related work.
> A: We will cite more recent works on lifelong learning in the related work section in the updated draft.
>
> Q5: Progressive networks increase in memory super-linearly?
> A: It is certainly true that we can add fewer units for new tasks due to forward transfer. The statement is based on the idea of original work on progressive networks [3] which initialize a new network each time for each new task.
>
>
> [1] Lopez-Paz, David, and Marc'Aurelio Ranzato. "Gradient episodic memory for continual learning." NeurIPS 2017.
> [2] Chaudhry, Arslan, et al. "Efficient lifelong learning with a-gem." ICLR 2019.
> [3] Rusu, Andrei A., et al. "Progressive neural networks." arXiv preprint arXiv:1606.04671 (2016).

---

> > ### Comment · AnonReviewer3 · 2019-11-14
> > **Could you provide some further clarifications?**
> >
> > I thank the authors for their response; there are still a few questions remaining.
> >
> > - With the direct approach, it seems odd to specify a loss threshold of zero to determine that the current task performance is high. What loss is being used?
> >
> > To clarify this question; what specific form does the loss take, such that a threshold of zero makes sense to gauge high performance? Eg. a mean-squared error loss would be strictly non-negative.
> >
> >
> > "A: It is certainly true that we can add fewer units for new tasks due to forward transfer. The statement is based on the idea of original work on progressive networks [3] which initialize a new network each time for each new task."
> >
> > Even in the case of a new network of equal size for each task, this would be linear; so I'm still not sure how the super-linearity claim can be justified.

---

> > > ### Author Response · Authors · 2019-11-14
> > > **Thanks for your responses! We provide more clarifications below.**
> > >
> > > - With the direct approach, it seems odd to specify a loss threshold of zero to determine that the current task performance is high. What loss is being used?
> > >
> > > A: We use the cross-entropy loss calculated on the batch of the current task. Since the training is online, the current batch can be viewed as a random sampled batch from the distribution of the current task [1], the loss on the current batch  can be seen as the generalization error of the current model. So a loss of zero can serve as a good indicator that the current task performance is high.
> > >
> > > - A: It is certainly true that we can add fewer units for new tasks due to forward transfer. The statement is based on the idea of original work on progressive networks [3] which initialize a new network each time for each new task."
> > >
> > > Yeas, if the network is of equal size for each task, then it's linear. However, in progressive networks, there are additional adapters or lateral connections as called in the paper to connect the new network with old ones. So the growth is much more than linear. The authors of Progressive Neural Network also pointed out this in Appendix B of the paper [2].
> > >
> > > [1] Nicolo Cesa-Bianchi, Alex Conconi, and Claudio Gentile. On the generalization ability of on-line
> > > learning algorithms. IEEE Transactions on Information Theory, 50(9):2050–2057, 2004.
> > >
> > > [2] Rusu, Andrei A., et al. "Progressive neural networks." arXiv preprint arXiv:1606.04671 (2016).

---

### Official Review · AnonReviewer2 · 2019-10-23
**Official Blind Review #2**

**Rating:** 3

**Review:**

========================== Summary of paper ==========================
The paper proposes a new loss balancing approach for lifelong learning. The proposed method dynamically balances the old task loss (from an episodic memory) and the new task loss based on their magnitude. The paper outperforms state-of-the-art method (that don't use extra attribute info), and is straightforward to understand.

========================== Decision ==========================
I vote for rejecting the paper. The paper adds a very simple dynamic loss balancing to a joint loss, which has limited novelty, yet does not discuss its relationship with loss balancing in multitask learning. Although the method outperforms state-of-the-art, a very simple baseline of their method also outperforms state-of-the-art, making the contribution ineffective. The writing of the paper may also benefit from further edits.

========================== Pros/Cons ==========================
+ The paper outperforms state-of-the-art.
+ The method is simple to explain and straightforward to implement.
+ Proposes a weighted variant of forgetting measure (although the purpose is not justified -- need to discuss what use case would make one prefer this weighted variant better than the original).
- The paper is not well-placed in the literature. Not until the bottom of page 5 can readers see very closely related methods, and despite the similarity, little is discussed about the difference. To improve, relationships with existing multitask learning weight balancing methods (e.g. https://arxiv.org/abs/1810.04650, https://arxiv.org/abs/1705.07115, https://arxiv.org/abs/1711.02257) should also be discussed, and maybe compared to.
- Motivation to use the proposed loss balancing rather than that of very similar methods (e.g. GEM/A-GEM) is lacking ("These approaches cannot capture the dynamics in the lifelong learning process", but GEM's balancing is also dynamic)
- Solving for beta using optimization (eq. 10), while (eq. 9) should have an explicit, close-form solution.
- Writing sometimes is confusing, uses absolute language, or using claims unsupported by evidence.
    - (1) page 3 "In this case, the weights are only optimized for the current task while ignoring previous tasks which leads to catastrophic forgetting." ignores the existence of regularization-based methods such as EWC
    - (2) bottom of page 3 "alpha1 and alpha2 should be adjusted adaptively" does not have experiment results supporting it (especially considering GEM is also adaptive)
    - (3) "l_ref(w; ζ) = 0 implies that there is almost no catastrophic forgetting" claim is problematic. Overfitting to the episodic memory is a common problem.
    - (3) page 3 xi and zeta not clearly defined.
    - (4) brings up NP-hardness while it is seldom of interest in this field.
    - (5) missing "∇" on the denominator in (eq. 7).
- Experiment:
    - The only ablation (the direct approach) is statistically indistinguishable from the proposed method. This also outperforms state-of-the-art, while it should not be. One can only assume that the experiment is problematic.
    - Completely uses hyperparameter from an unpublished paper.
    - Claims state-of-the-art, yet omits a state-of-the-art variant in cited A-GEM paper (with joint-embedding). This should be discussed even if the comparison is unfair.

========================== Improving the paper ==========================
- Rewrite the introduction so it is clear the paper is an improvement over A-GEM with a better loss balancing, and focus the motivation and side experiments on why this is important for lifelong learning.
- Clean up writing.
- Explain why the direct approach, while being a basic loss balancing method, outperform state-of-the-art GEM greatly. Perhaps a set of ablation studies can help.
- Replace optimizing for beta as solving for beta.
- Comparison with other loss balancing papers.
- Clarify state-of-the-art comparison.

========================== After rebuttal ==========================
I appreciate the authors addressing my concerns about the placement in the literature; it looks clearer now. However, the rebuttal did not address well the question why the simple MEGA-D variant outperforms prior state-of-the-art besides promising to publish code. The new ablation study is especially difficult to understand, since the first set of results are MEGA without using both memory and new data, but it seems to me that MEGA *needs* the memory data to work, so I have no idea what is being ablated here, and I also do not understand how this answers the question why MEGA-D performs so well. I will not be changing my rating.

Other issues:
(1) Citing A-GEM: please update the bib file so it shows the ICLR version instead of arXiv.
(2) Solving eq. 9: Here you go: https://imgur.com/a/b4QCVlv
I literally had this problem on one of my high school exams. Feel free to use this in any future version.

**Experience Assessment:**

I have published one or two papers in this area.

**Review Assessment: Checking Correctness Of Derivations And Theory:**

I carefully checked the derivations and theory.

**Review Assessment: Checking Correctness Of Experiments:**

I did not assess the experiments.

**Review Assessment: Thoroughness In Paper Reading:**

I read the paper at least twice and used my best judgement in assessing the paper.

---

> ### Author Response · Authors · 2019-11-10
> **Reply to Reviewer 2 (Part 1)**
>
> We greatly appreciate the time and effort of the reviewer to point out the issues of the current draft. We will incorporate the comments in the final manuscript to improve the quality and readability of the paper.
>
> Q1: The purpose the weighted variant of forgetting measure
>
> A:  The proposed LTR measures if the method can maintain its performance on the tasks trained initially. The purpose of the proposed measure is to access the accuracy drop on each task during the learning process.
>
> Q2: Not well-placed in the literature.
>
> A: Thanks for pointing out this. We will add more discussions on related works in the beginning of the draft to clearly show the differences between the proposed method and existing methods to improve the readability of the paper.
>
> Q3: Motivation to use the proposed loss balancing rather than that of very similar methods (e.g. GEM/A-GEM).
>
> A:  Thanks for the question. We agree that the sentence “these approaches cannot capture the dynamics” is not very accurate, what we want to emphasize is that we explicitly consider the loss on the current task and old tasks to achieve a better balance.  Below we add discussions on the motivations of the proposed approach and explain why our method can achieve higher accuracy than previous approaches.
>
> First, we explicitly maximize the correlation between the mixed stochastic gradient g_mix and the current gradient g_cur (calculated on current data), and the correlation between the mixed stochastic gradient g_mix and the reference gradient g_ref (calculated on memory data) as in Eq (7). Intuitively, the direction of g_mix will not bias towards the current task or old tasks  (since it is easy to see that the angle between g_mix and g_cur, and the angle between g_mix and g_ref are both acute except the edge case where the angle between g_cur and g_ref is 180 degree). By following g_mix, each update tends to degrade the loss on both current task and old tasks.   While in previous works (GEM [1], A-GEM[2]), the corresponding g_mix is found by projecting the current gradient to be perpendicular to g_ref when angle between them is obtuse (in this case, g_mix $\cdot$ g_ref = 0). Intuitively, both GEM and A-GEM put more emphasis on the current task (when g_mix is perpendicular to g_ref, the update following the direction of g_mix will not directly reduce the loss on old tasks [1]). Although GEM and A-GEM also adaptively modify the direction of the current gradient, our approach can achieve a better tradeoff between the performance on current task  and old tasks based on above reasoning.
>
>  We did an ablation study on just adding this source,
> ############################################################################
> The results of only considering the first source (maximizing correlation):
> On MNIST:    91.15			On CIFAR:   58.04
> On CUB:       68.6			        On AWA:      47.95
>
> ############################################################################
> Results of A-GEM [2]:
>
> On MNIST:    89.32				On CIFAR:   61.28
> On CUB:        61.82				On AWA:      44.95
> ############################################################################
>
> Second, we introduce the loss on the current batch and on the sampled batch from episodic memory. This allows us to better balance the performance on the current task and old tasks. Since it is preferable to put more emphasis on the tasks (task) which achieve(s) low performance.
> ############################################################################
> The results of combining two sources:
> On MNIST:  91.21  			On CIFAR: 66.12
> On CUB:     80.58			        On AWA:    54.28
> ############################################################################
>
> It can seen that both of the two sources contribute to the improvement.  We will add the ablations and discussions of each source in the final draft to clearly demonstrate why our approach is better than previous approaches.
>
> In summary, we first propose a better approach to adaptively modify the direction of the gradient.  Second, we introduce the losses on current task and on the episodic memory to achieve a better balancing.  Hopefully the visualization below can better explain the relation between GEM/A-GEM/MER, MEGA and MEGA-D,
>
> GEM/A-GEM/MER $\rightarrow^1$ MEGA $\rightarrow^2$ MEGA-D
>
> 1. Better modification of the current gradient direction by maximizing correlation as in Eq 7 and better balancing with losses on current task and old tasks.
>
> 2. Simple variant of MEGA which balances current task and old tasks by only leveraging loss information.

---

> ### Author Response · Authors · 2019-11-10
> **Reply to Reviewer 2 (Part 2)**
>
> Q4: Relationship with existing multitask learning weight balancing methods should also be discussed.
>
> A: Thanks for pointing out this. We will include the references you mentioned and discuss them in the revised version. We also want to highlight the difference between our approach and the approaches in multitask learning. Lifelong learning is an online learning paradigm while multi-task learning is offline. Multi-task learning aims at optimizing the network to produce multiple predictive outputs for different tasks on the same dataset. However in lifelong learning, each task is a dataset. The most closely related work in the mentioned reference is GradNorm https://arxiv.org/abs/1711.02257, the main idea of GradNorm is to balance tasks such as depth estimation and segmentation based on the magnitude of the gradients, however in the proposed MEGA and also in GEM [1], we focus on modifying the gradient direction to overcome catastrophic forgetting in lifelong learning.
>
>
> Q5: Solving for beta using optimization (eq. 10), while (eq. 9) should have an explicit, close-form solution.
>
> A: The objective function of (eq. 9) is a possibly non-convex function in terms of $\beta$. It is impossible to get a general closed-form solution to the best of our knowledge. The best we can do is to derive closed-form solution if $\ell_t=\ell_{\text{ref}}$ since in this case the function is concave. For example, the objective of (8) is concave and the optimal solution of $\beta$ is $\tilde{\theta}/2$.
>
> Q6: Writing sometimes is confusing.
>
> (1) This sentence is only applied to the optimization problem formulated in Eq 4. And we did not intend that it can be generalized to other type of lifelong learning methods.
>
> (2) “bottom of page 3 "alpha1 and alpha2 should be adjusted adaptively" does not have experiment results supporting it (especially considering GEM is also adaptive)”
>
> A: Please refer to the answer of Q3 in Part 1.
>
> (3) “ "l_ref(w; ζ) = 0 implies that there is almost no catastrophic forgetting" claim is problematic. Overfitting to the episodic memory is a common problem ”
>
> A: We use l_ref(w; ζ) = 0 as a proxy to assess the performance of the current model on old tasks. We agree that there can be overfitting when the memory size is small. Since we only compute the loss on a sampled batch from the episodic memory, this can alleviate the overfitting issue to some extent. Empirically we found the loss on the episodic memory can still serve as a good indicator to balance old tasks and the current task as shown in the ablation studies in the answer to Q3 in Part 1.
>
> (4) “page 3 xi and zeta not clearly defined.”
>
> A:  xi and zeta are random variables which characterize the random samples. This is a commonly usage in stochastic optimization [3].
>
> (5) “brings up NP-hardness while it is seldom of interest in this field.”
>
> A: We use NP-hard to motivate the necessity to use stochastic gradient descent to solve the problem. We will remove this in the updated draft.
>
> (6) “missing "∇" on the denominator in (eq. 7).”
>
> A: Thanks for pointing out this. We will make a correction in the updated draft.
>
>
> Q7: Experiments
>
> (1) The only ablation (the direct approach) is statistically indistinguishable from the proposed method. This also outperforms state-of-the-art, while it should not be. One can only assume that the experiment is problematic.
>
> A: The direct approach is a variant of the MEGA (can also be regarded as an ablation as the reviewer pointed out) which can be easily implemented. Our experiments follow the exact same setting as the previous work [2] for a fair comparison. We open source our code and the results can be easily replicated.  If there is any incorrectness in the experiment, we welcome the reviewer to point out.
>
> (2) Completely uses hyperparameter from an unpublished paper.
>
> A: We reuse the hyperparameters from A-GEM [2] which is published in ICLR 2019.
>
> (3) Claims state-of-the-art, yet omits a state-of-the-art variant in cited A-GEM paper (with joint-embedding).
>
> A: The joint-embedding utilizes attribute information.  For the state-of-the-art comparison, we only consider the label of the examples without any other information. This is applied to all the methods considered in the paper to allow a fair comparison.
>
> [1] Lopez-Paz, David, and Marc'Aurelio Ranzato. "Gradient episodic memory for continual learning." NeurIPS 2017.
> [2] Chaudhry, Arslan, et al. "Efficient lifelong learning with a-gem." ICLR 2019.
> [3] Johnson, Rie, and Tong Zhang. "Accelerating stochastic gradient descent using predictive variance reduction." Advances in neural information processing systems. 2013.

---

### Official Review · AnonReviewer1 · 2019-10-24
**Official Blind Review #1**

**Rating:** 3

**Review:**

I am very torn about this paper as the experiments are thorough and the results are quite significant. The results are on four popular benchmarks and they also test on Many Permutations, which is an important finding as well.  While the angle based approach presented here is interesting and intuitively appealing, I am not really sure whether it is a needed generalization beyond past approaches. I don't really have a strong intuition for where the gains are coming from. As such, I think the paper would be much stronger if it could intuitively or theoretically get at why it improves on past approaches.

Actually, one of my biggest concerns is how good the performance of the direct approach is. The direct approach seems like a straightforward extension of experience replay with a special learning rate for current examples vs. buffer examples that is adaptive to the relative loss.  While it is potentially interesting that this approach works so well, it is not positioned as the selling point of the paper as currently written. Additionally, this is kind of an orthogonal contribution to past papers leveraging experience replay, which could potentially be modified and improved based on this approach. I wonder if the authors tried modifying any of the baseline approaches in this way as well.

A small note on the experiments: the authors closely follow settings from past work and at times it can be unclear which results were implemented in this paper and which in past papers. I don't think that you have addressed what hyperparameters you chose for MER. It also seems to perform worse than past reported results on CIFAR and is not reported despite the best past performance on MNIST Permutations.

**Experience Assessment:**

I have published in this field for several years.

**Review Assessment: Checking Correctness Of Derivations And Theory:**

I assessed the sensibility of the derivations and theory.

**Review Assessment: Checking Correctness Of Experiments:**

I assessed the sensibility of the experiments.

**Review Assessment: Thoroughness In Paper Reading:**

I read the paper at least twice and used my best judgement in assessing the paper.

---

> ### Author Response · Authors · 2019-11-10
> **Reply to Review 1 (Part 1)**
>
> We thank the reviewer for your valuable comments and insightful questions. Next, we answer  the questions in detail.
>
> Q1: Where are the gains are coming from?
>
> A:  Thanks for the question. We definitely agree that this is an important question which has not been addressed well in the current draft.  In summary, the benefits of our approach come from two sources.
>
> First, we explicitly maximize the correlation between the mixed stochastic gradient g_mix and the current gradient g_cur (calculated on current data), and the correlation between the mixed stochastic gradient g_mix and the reference gradient g_ref (calculated on memory data) as in Eq (7). Intuitively, the direction of g_mix will not bias towards the current task or old tasks  (since it is easy to see that the angle between g_mix and g_cur, and the angle between g_mix and g_ref are both acute except the edge case where the angle between g_cur and g_ref is 180 degree). By following g_mix, each update tends to degrade the loss on both current task and old tasks.   While in previous works (GEM [1], A-GEM[2]), the corresponding g_mix is found by projecting the current gradient to be perpendicular to g_ref when angle between them is obtuse (in this case, g_mix $\cdot$ g_ref = 0). Intuitively, both GEM and A-GEM put more emphasis on the current task (when g_mix is perpendicular to g_ref, the update following the direction of g_mix will not directly reduce the loss on old tasks [1]).  We did an ablation study on just adding this source,
>
> ############################################################################
> The results of only considering the first source (maximizing correlation):
> On MNIST:    91.15			On CIFAR:   58.04
> On CUB:       68.6			On AWA:      47.95
>
> ############################################################################
> Results of A-GEM [2]:
>
> On MNIST:    89.32				On CIFAR:   61.28
> On CUB:       61.82				On AWA:      44.95
> ############################################################################
>
> Second, we introduce the loss on the current batch and on the sampled batch from episodic memory. This allows us to better balance the performance on the current task and old tasks. Since it is preferable to put more emphasis on the tasks (task) which achieve(s) lower performance.
>
> ############################################################################
> The results of combining two sources:
> On MNIST:  91.21  			On CIFAR: 66.12
> On CUB:     80.58			        On AWA:    54.28
> ############################################################################
>
> It can seen that both of the two sources contribute to the improvement.  We will add the ablations and discussions of each source in the final draft to clearly demonstrate why our approach is better than previous approaches.
>
>
> Q2: The performance of the direct approach.
> A:  Thanks for the question.
>
> We need to point out that both MEGA and MEGA-D are our contributions and previous works such as GEM [1] and A-GEM [2] did not propose similar approaches.  For MEGA-D, our intention is to provide a simple variant of MEGA which can be easily implemented and can serve as an off-the-shelf solution for lifelong learning tasks.  For more difficult lifelong learning tasks, we found MEGA still performs better than MEGA-D. We did an additional experiment on Many Permutations on MNIST (100 tasks and 200 examples per task) below,
>
> Average accuracy over 5 five runs on Many Permutations with MNIST (100 tasks and 200 examples per task) :
>
> MEGA-D: 56.52 ± 0.43		MEGA: 62.48 ± 0.51
>
> We will include the updated results in the draft.
>
>
> Q3: Experiments.
> A: We use the public code released by [2] to produce all the results if the method is available in the code, and our results match the results reported in [2].  For MER, we also adopt the code published by the original author. Unfortunately, we cannot find the results on CIFAR in the original MER paper.  To obtain results of MER on CIFAR, we reuse the hyperparameters of A-GEM [2], the same hyperparameters are also applied to our methods.  For MNIST Permutations, MER adopt a different setting from [2] (MER) which only consider 1000 examples per task and their best result is 85.50. We tried to run MER on our setting where each task has 60000 examples, but unfortunately MER cannot scale to such sample size since the model is trained one example at a time based on the meta-learning procedure. The best we can do is to run MER on Many Permutations (100 tasks and 200 examples per task) for MNIST.
>
> [1] Lopez-Paz, David, and Marc'Aurelio Ranzato. "Gradient episodic memory for continual learning." NeurIPS 2017.
> [2] Chaudhry, Arslan, et al. "Efficient lifelong learning with a-gem." ICLR 2019.

---

> > ### Author Response · Authors · 2019-11-10
> > **Reply to Review 1 (Part 2)**
> >
> > Hope the illustration below can better explain the relation between GEM/A-GEM/MER, MEGA and MEGA-D
> >
> >
> > GEM/A-GEM/MER $\rightarrow^1$ MEGA $\rightarrow^2$ MEGA-D
> >
> > 1. Better modification of the current gradient direction by maximizing correlation as in Eq 7 and better balancing with losses on current task and old tasks.
> >
> > 2. Simple variant of MEGA which balances current task and old tasks by only leveraging loss information.
> >
> >
> > More clarifications of MEGA-D and A-GEM/GEM:
> >
> > we want to further point out that it is not clear how to directly apply MEGA-D to A-GEM/GEM, i.e., introduce the loss information to A-GEM/GEM.  Since the gradient direction modification rule of A-GEM/GEM in some sense is "hard-coded".
> >
> > In A-GEM/GEM, if the angle between the current gradient and the reference gradient is acute, then there is no change to the current gradient. On the other hand, if the angle between the current gradient and the reference gradient is obtuse, then project the current gradient to be perpendicular to the reference gradient to the obtain the projected gradient.  In both cases,  we cannot directly multiply the loss (or the ratio of the current loss and reference loss as in the direct approach)  to the current gradient or the projected gradient, since the loss (or the ratio of the current loss and reference loss as in the direct approach)  can only change the magnitude of the current gradient or the projected gradient but not the direction.  In MEGA-D, the final gradient is a linear combination of the current gradient and the reference gradient, so it is very natural to use loss to balance the two gradients. The direction of the final gradient obtained by MEGA-D can vary based on the ratio of the current loss and reference loss.

---

### Author Response · Authors · 2019-11-10
**Updated version of the paper is uploaded**

Dear Reviewers,

We have incorporated all the suggestions and comments into the updated draft.  We add more related works and additional experiments. Due to the space limitation, we add the ablation studies in the Appendix. Please let us know if you have more suggestions or comments. We are happy to include them to improve the quality and readability of the paper.

Best,

Paper318 Authors

---

### Decision · Program_Chairs · 2019-12-19

**Decision:**

Reject

**Comment:**

The submission is concerned with the catastrophic forgetting problem of continual learning, and proposes a gradient-based method which uses buffers of data seen previously to integrate the angles of the gradients and thereby mitigate forgetting. Empirical results are given on several benchmarks.

The reviewers were impressed with the thorough validation and strong results, but noticed that the much simpler MEGA-D baseline did almost as well. Given this, they were not convinced that the proposed approach was necessary. Although the authors provided a strong rebuttal and an additional ablation, the reviewers did not feel that their concerns were met.

My recommendation is to reject the submission at this time.